DOI: 10.1038/s41467-018-04818-0　　**OPEN**

# Uhrf1 regulates active transcriptional marks at bivalent domains in pluripotent stem cells through Setd1a

Kun-Yong Kim[1], Yoshiaki Tanaka[1], Juan Su[1,2], Bilal Cakir [1], Yangfei Xiang[1], Benjamin Patterson[1], Junjun Ding[3], Yong-Wook Jung[4], Ji-Hyun Kim[1], Eriona Hysolli[1], Haelim Lee[1], Rana Dajani[5,6], Jonghwan Kim [7], Mei Zhong[8], Jeong-Heon Lee[9], David Skalnik[9], Jeong Mook Lim[10], Gareth J. Sullivan[11,12], Jianlong Wang [3] & In-Hyun Park[1]

Embryonic stem cells (ESCs) maintain pluripotency through unique epigenetic states. When ESCs commit to a specific lineage, epigenetic changes in histones and DNA accompany the transition to specialized cell types. Investigating how epigenetic regulation controls lineage specification is critical in order to generate the required cell types for clinical applications. Uhrf1 is a widely known hemi-methylated DNA-binding protein, playing a role in DNA methylation through the recruitment of Dnmt1 and in heterochromatin formation alongside G9a, Trim28, and HDACs. Although Uhrf1 is not essential in ESC self-renewal, it remains elusive how Uhrf1 regulates cell specification. Here we report that Uhrf1 forms a complex with the active trithorax group, the Setd1a/COMPASS complex, to maintain bivalent histone marks, particularly those associated with neuroectoderm and mesoderm specification. Overall, our data demonstrate that Uhrf1 safeguards proper differentiation via bivalent histone modifications.

[1] Department of Genetics, Yale Stem Cell Center, Yale Child Study Center, Yale School of Medicine, New Haven, CT 06520, USA. [2] Department of Cell Biology, the Second Military Medical University, 200433 Shanghai, China. [3] Department of Cell, Developmental and Regenerative Biology, Black Family Stem Cell Institute, Icahn School of Medicine at Mount Sinai, New York, NY 10029, USA. [4] Department of Obstetrics and Gynecology, CHA Gangnam Medical Center, CHA University, Seoul 06135, Republic of Korea. [5] Department of Biology and Biotechnology, Hashemite University, Zarqa 13115, Jordan. [6] Radcliffe Institute for Advanced Studies, Harvard University, Cambridge, 02138 MA, USA. [7] Department of Molecular Biosciences, Institute for Cellular and Molecular Biology, Center for Systems and Synthetic Biology, the University of Texas at Austin, Austin, TX 78712, USA. [8] Department of Cell Biology, Yale Stem Cell Center, Yale School of Medicine, New Haven, CT 06520, USA. [9] Department of Biology, School of Science, Indiana University-Purdue University Indianapolis, Indianapolis, IN 46202, USA. [10] Department of Agricultural Biotechnology, Seoul National University, Seoul 08826, Korea. [11] Department of Molecular Medicine, Hybrid Technology Hub - Centre of Excellence, Institute of Basic Medical Sciences, University of Oslo, 0372 Oslo, Norway. [12] Norwegian Center for Stem Cell Research, Institute of Immunology, Oslo University Hospital and University of Oslo, 0372 Oslo, Norway. These authors contributed equally: Kun-Yong Kim, Yoshiaki Tanaka. Correspondence and requests for materials should be addressed to I.-H.P. (email: inhyun.park@yale.edu)

Uhrf1 (Ubiquitin-like, with PHD and RING finger domains 1, also known as NP95 or ICBP90) is a multi-domain nuclear protein that faithfully regulates epigenetic modifications through two mechanisms: (i) by recognition of histone marks through subsequent interactions with chromatin modifying proteins and (ii) DNA methylation maintenance[1]. Uhrf1 is essential in early embryogenesis[2–4]. Although Uhrf1 knock-out (KO) mouse embryonic stem cells (ESCs) are viable and able to self-renew, they display delayed cell cycle progression, a loss of DNA methylation, altered chromatin structure, and enhanced transcription of repetitive elements[2,4]. Uhrf1 is also highly expressed in neural stem cells (NSCs). Interestingly, loss of Uhrf1 in NSCs leads to the activation of retroviral elements, similar to that observed in Uhrf1 KO ESCs[5]. Recent studies showed that a reduction of Uhrf1 expression via Pramel7 (PRAME-like 7) is important in the conversion of primed ESCs to a naive state[6,7].

One of the major functions of Uhrf1 is the inheritance of DNA methylation during DNA replication. Uhrf1 binds to hemimethylated DNA via its Set- and RING-Associated (SRA) domain, which facilitates the loading of DNA methyltransferase 1 (Dnmt1) onto the newly synthesized DNA strand during cell division[8]. The plant homeo domain (PHD) and tandem Tudor domain (TTD) domains of Uhrf1 simultaneously recognize trimethylated H3 at lysine 9 (H3K9me3), which could potentially contribute to the interplay between histone modification and DNA methylation, and the localization of H3K9me3 to pericentric heterochromatin[9–11]. Uhrf1 also contains a really interesting new gene (RING) domain that ubiquitylates histone H3 at lysine 23 (H3K23ub) and is essential for the recruitment of Dnmt1 for the maintenance of DNA methylation[12]. Recent discoveries have demonstrated Uhrf1's bipartite role as a DNA damage sensor and nuclease scaffold in DNA repair, as well as the importance of its SRA domain[13–15]. Although the biochemical function of Uhrf1 in DNA methylation and heterochromatin formation has been extensively investigated, its biological function in ESCs has yet to be explored.

Bivalent histone marks, represented by H3K4me3 and H3K27me3, are unique features of promoters associated with development and differentiation in ESCs[16]. When ESCs differentiate into a given lineage, active histone marks are maintained in genes that are expressed in that specific lineage, while the repressive histone marks in those genes are concomitantly removed[16]. The polycomb repressive complex 2 (PRC2) proteins mediate H3K27me3 modification to regulate gene repression[17,18]. In contrast, H3K4 methylation is catalyzed by the Set1 complex proteins. Metazoans have three subsets of this complex: the Set1/COMPASS, trithorax (Trx), and trithorax-related (Trr). These complexes share the same core protein components, but differ in their catalytic subunits. The Set1/COMPASS complex has Setd1a or Setd1b as its catalytic subunit, while Trx has myeloid/lymphoid or mixed-lineage leukemia 1 (MLL1) or MLL2, and Trr has MLL3 or MLL4[19]. Set/MLL core subunits, such as WD repeat-containing protein 5 (Wdr5), Ash2l (Ash2-like), and retinoblastoma-binding protein 5 (Rbbp5), are required for full histone methyltransferases (HMT) activity of the Set complex, while Rbbp5 and Ash2l heterodimer participates in the HMT activity of MLL1 complex[20–23]. In spite of overwhelming evidence that Uhrf1 regulates repressive histone marks, it is still unclear whether Uhrf1 is involved in the regulation of active chromatin marks.

Here, we investigate the function of Uhrf1 in its regulation of pluripotency and differentiation of ESCs. Surprisingly, our data show that Uhrf1 plays a critical role in lineage specification by controlling bivalent histone modifications. Its deletion in ESCs disrupts not only the repressive mark H3K27me3, but also the active histone mark H3K4me3 on bivalent loci, ultimately causing defects in differentiation. Furthermore, biochemical analysis demonstrates that Uhrf1 interacts with the Setd1a/COMPASS complex and positively regulates H3K4me3 modifications. Our findings reveal an essential function of Uhrf1 as a stabilizer of the epigenome by promoting H3K4me3 modifications necessary for faithful differentiation and the maintenance of bivalent histone modifications for pluripotency.

## Results

**Uhrf1 deficiency disrupts bivalent histone marks in ESCs.** We first performed chromatin-immunoprecipitation with high-throughput sequencing (ChIP-seq) to identify global targets of Uhrf1. 2784 Uhrf1-enriched regions (10.2%) were identified around promoters or gene bodies, while 10,860 were located in the intergenic regions (89.8%) (Fig.1a). Comparative analysis with ChIP-seq for histone modifications showed that the genome-wide distribution of Uhrf1 is most significantly correlated with that of H3K9me3 marks (Pearson correlation = 0.928, $p < 2.2e{-}16$, Supplementary Fig. 1a), consistent with previous findings that Uhrf1 directly interacts with H3K9me3[24]. Interestingly, Uhrf1 distribution was also highly correlated with H3K4me3 and H3K27me3 mark distribution (correlation = 0.582 and 0.747, respectively) (Supplementary Fig. 1a). In contrast, 5-hydroxymethyl-cytosine (5hmC) or methyl-cytosine (5mC) did not appear to correlate with Uhrf1-binding patterns (correlation = −0.113 and −0.149, respectively) (Supplementary Fig. 1a)[25]. In addition, comparative analysis of transcriptome profiles of Uhrf1 KO and Dnmt1 KO ESCs showed that only 12.3% of the up-regulated genes and 4.4% of the down-regulated genes overlapped (Fig. 1b)[26]. Pluripotent-related genes (e.g. *Prdm14*, *Rex1* and *Lefty1*) were dysregulated uniquely in Dnmt1 KO ESCs, while Uhrf1 KO cells exhibited dysregulation in several developmental genes (Hand1 and Sox15), and genes exclusively expressed at the early embryonic stage (e.g. *Klf2* and *Zscan4b*) (Fig. 1b). Gene ontology (GO) analysis also revealed that genes involved in protein ubiquitination and DNA repair, the known functions of Uhrf1, were uniquely down regulated in Uhrf1 KO ESCs[14,25]. These results suggest that Uhrf1 chromatin occupancy is highly associated with histone modification rather than DNA methylation, despite the fact that Uhrf1 is involved in the recruitment of Dnmt1 to facilitate DNA methylation through H3K9me3 binding[10].

To determine Uhrf1's contribution to chromatin organization, we compared various histone modifications (H3K4me3, H3K27me3, and H3K9me3) between wild type (WT) and Uhrf1 KO ESCs. We found that Uhrf1 deficiency disrupted H3K4me3, H3K27me3, and H3K9me3 modifications (Supplementary Fig. 1b). Notably, disruption of H3K4me3 and H3K27me3 preferentially occurred at transcription start sites (TSSs) ($p < 2.2e{-}16$, "E" in Fig. 1c), whereas H3K9me3 disruptions occurred predominantly in intergenic regions (Supplementary Fig. 1c), L1 and LTR class retrotransposons (Supplementary Fig. 1d). Our results suggest that Uhrf1 not only localizes to heterochromatic regions, but also to specific euchromatic regions for transcriptional regulation of genes through regulation of H3K4me3 and H3K27me3 modifications.

Using WT and Uhrf1 KO ESCs, we next classified promoters into four distinct states, either active, repressive, bivalent, or no mark, indicated by the presence or absence of H3K4me3 and H3K27me3 marks (Supplementary Fig. 1e). We observed that while the majority of active and no-mark promoters were retained in Uhrf1KO ESCs, half of repressive promoters (50.4%) were switched to no-mark promoters in Uhrf1KO ESCs. Interestingly, 55% of the bivalent marks observed in WT ESCs

were disrupted and redistributed into either repressive (13.95%) or active (33.57%) promoter states in Uhrf1 KO cells (Fig. 1d). Uhrf1 was significantly enriched around bivalent and repressive loci ($p < 3.07e-2$, Supplementary Fig. 1f). In addition, Uhrf1 KO ESCs showed a significant decrease in H3K4me3 and H3K27me3 modifications at the promoters of mesodermal genes and a

decrease of H3K4me3 in ectodermal genes ($p < 4.86-2$ by T test, Supplementary Fig. 1g). This was further validated using ChIP-qPCR, against genes involved in differentiation and pluripotency. For example, genes involved in muscle development (e.g. *Msx2* and *Hand2*) showed a marked reduction of H3K4me3 and H3K27me3, while neuroectoderm (e.g. *Tubb3* and *Nes*) and

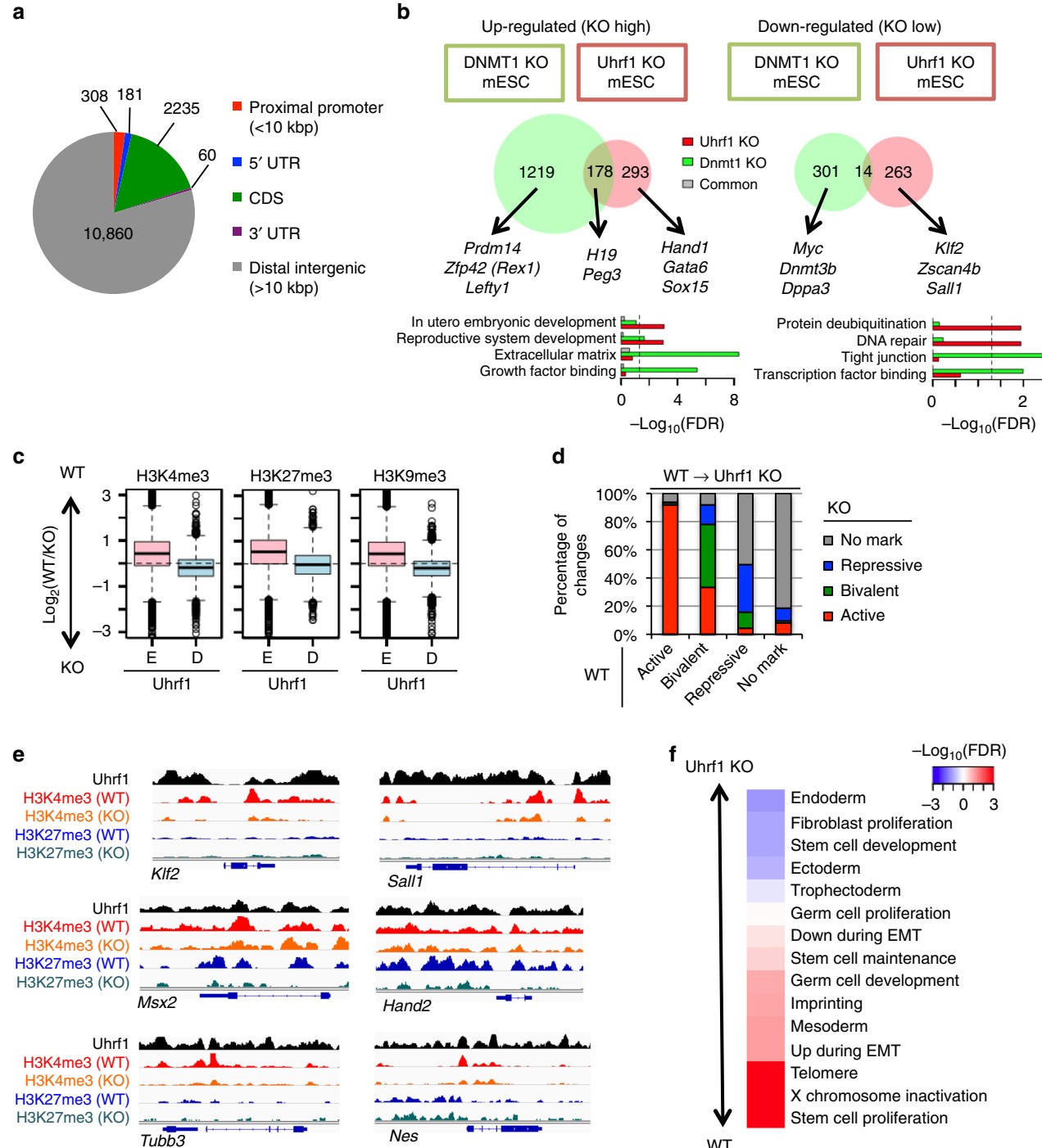

**Fig. 1** Uhrf1 deficiency disrupts bivalent histone modification. **a** Distribution of Uhrf1-enriched regions in gene body and intergenic regions. **b** Comparative analysis of up-regulated (left) and down-regulated genes (right) between Uhrf1 and Dnmt1 KO. Representative genes and over represented GO terms for unique and common genes are also shown. Dashed line represents 0.05 FDR cut-off. **c** Comparative analysis of H3K4me3, H3K27me3, and H3K9me3 change between Uhrf1-enriched (**e**) and Uhrf1-depleted (**d**) regions. Y-axis represents the difference of log₂(ChIP/input) between WT and Uhrf1 KO. **d** Changes of promoter classes between WT and Uhrf1 KO ESCs. **e** Histone modification landscape of representative genes related to pluripotency, mesoderm/mesenchyme/muscle, and neuroectoderm development. **f** GSEA for stem cell functions. Statistical enrichment (−log₁₀(FDR)) in Uhrf1 KO and WT was shown by blue and red color, respectively

pluripotency genes (e.g. *Klf2* and *Sall1*) displayed a loss of H3K4me3 (Fig. 1e and Supplementary Fig. 1h, i).

To determine whether the disruption of a bivalent state in poised promoters by KO of Uhrf1 truly affects gene expression levels, we compared global gene expression between WT and Uhrf1 KO ESCs. Surprisingly, only a few hundred genes were differentially expressed between WT and Uhrf1 KO ESCs (Fig. 1b). Despite the observed differential modification of H3K4me3 and H3K27me3 on the ectodermal and mesodermal genes (Supplementary Fig. 1g), Uhrf1 deficiency in ESCs did not significantly affect stem cell maintenance (FDR = 0.298), trilineage developmental genes (FDR > 0.060) or trophectoderm (FDR = 0.501) (Fig. 1f)[27]. Although Uhrf1-derived bivalent disruption had little influence on trilineage-related gene expression in Uhrf1 KO ESCs, disrupted bivalent marks in Uhrf1 KO ESCs reveal a previously undefined function of Uhrf1 as a regulator of cell lineage specification during differentiation.

**Uhrf1 is critical in mesodermal and ectodermal differentiations**. To address if bivalent disruption impacts ESCs differentiation, we assessed embryoid body (EB) formation using WT and Uhrf1 KO ESC lines. Uhrf1 KO EBs presented with an irregular and inflated shape as early as day 7 when compared to the spherical EBs derived from WT ESCs (Fig. 2a). In addition to the gross morphological differences, Uhrf1 KO ESCs formed contractile EBs earlier than WT ESCs. WT EBs started beating at around day 9 and showed a gradual decrease in beating until day 21, while Uhrf1 KO EBs exhibited rhythmical beating as early as day 5. The proportion of beating EBs in Uhrf1 KO dramatically decreased after day 10, and the majority of EBs had ceased to beat by day 12 (Fig. 2b). Gene expression analyses of trilineage markers showed an upregulation of the mesodermal markers *Nkx2.5* and *Tbx5* in the early stages of Uhrf1 KO EB formation (Fig. 2c). In contrast, the neuroectodermal markers *Sox1* and *TuJ* showed down regulation in Uhrf1 KO cells, compared with WT cells. The observed bias in differentiation of Uhrf1 KO ESCs towards mesoderm, combined with reduced neuroectoderm potential was further clarified by immunostaining against early lineage-specific markers (Fig. 2d). Moreover, when Uhrf1 KO ESCs were differentiated towards the neural lineage by exposure to retinoic acid (RA), we observed a striking increase of cell death and a failure to neuralise (Supplementary Fig. 2a). To eliminate potential clonal variation, we derived the following mESC cell lines: JES6 derived from Uhrf1^{fl/fl} mouse and the Uhrf1-null mESC line (JES6-Cre) (Supplementary Fig. 2b, c). Consistent with the already described Uhrf1 KO cell line, JES6-Cre mESCs presented with the same defects in both neural differentiation and expedited formation of beating EBs (Supplementary Fig. 2d, f). These results importantly point towards a previously unidentified role of Uhrf1 in regulating lineage specification.

To further evaluate the role of Uhrf1 in ESC differentiation in vivo, we performed a teratoma assay by intramuscular injection of Uhrf1 KO and WT ESCs into immune-deficient mice. We were able to isolate teratomas derived from Uhrf1 KO ESCs that contained representative tissues for all the three germ layers (Fig. 2e). However, teratomas were much smaller as compared to WT ESCs (Fig. 2f and Supplementary Fig. 2g). This highlighted that the observed differentiation defect exerted by Uhrf1 KO is a phenotype that occurs both in vitro and in vivo.

In order to test whether the observed differentiation defect was a consequence of the absence of Uhrf1 and not due to abnormalities inherent with Uhrf1 KO ESC line, we ectopically expressed recombinant Uhrf1 in Uhrf1 KO cells. Importantly, the introduction of Uhrf1 rescued the abnormal morphology in EBs (Fig. 2a—lower panel), as well as premature mesodermal

differentiation, as demonstrated by the recovery of EB beating time (Fig. 2b). Furthermore, the expression of trilineage markers, in particular the neuroectoderm markers *Sox1* and *TuJ*, were rescued to WT levels (Fig. 2c). The accelerated expression of mesodermal markers *Bmp4* and *Nkx2.5* at early stage of Uhrf1 KO EBs was rescued to near WT levels. Taken together, Uhrf1 regulates the expression of lineage-specific genes to achieve correctly mediated ESC differentiation.

To gain more global insight into the role of Uhrf1 in the regulation of differentiation, RNA-seq was performed with EBs from both WT and Uhrf1 KO cells. We observed a large number of genes that were differentially expressed as differentiation progressed (Supplementary Fig. 3a). Consistent with the contractile EB pattern, a transcriptional module of muscle and cardiomyocyte developmental-related genes showed higher expression in Uhrf1 KO EBs at week 1, but dramatically decreased at week 2 and week 3 as compared to WT EBs (module 23 in Fig. 2g). Consistent with qPCR results (Fig. 2c), a gene set related to ectodermal development (neural crest and neuron development) was down regulated in Uhrf1 KO EBs when compared with WT (module 26 in Fig. 2g).

To further investigate whether the observed higher mesodermal lineage differentiation and neuroectodermal commitment defects were related to disruption of bivalent chromatin on developmental promoters, we performed ChIP-seq against H3K4me3 and H3K27me3 in both WT and Uhrf1 KO EBs. We observed a similar pattern to undifferentiated ESCs with respect to the active or no-mark, as these were largely unchanged in Uhrf1 KO EBs (Supplementary Fig. 3b). However, both bivalent and repressive promoters became increasingly disrupted, concomitant with EB differentiation. GO analysis showed that the bivalent disruptions were enriched near promoters of lineage-specific genes (Fig. 2h). Interestingly, mesodermal and neuroectodermal lineage markers displayed different promoter modification defect in Uhrf1 KO cells. For example, neuroectodermal genes (*Edrna*, *Otx2*, and *Neurod1*) displayed a failure to gain of H3K4me3 marks on their promoter regions during Uhrf1 KO EB differentiation (FDR < 8.61e−3), while the muscle or heart developmental genes (*Nkx2.5*, *Tbx5*, and *Wnt2b*) showed a loss of the H3K27me3 modification on bivalent promoters (Supplementary Fig. 3c). These differential depositions of histone marks on two distinct lineages may account for the biased differentiation observed in Uhrf1 KO ESCs.

**Uhrf1 regulates the H3K4me3 with Setd1a/COMPASS complex**. Next, we asked how Uhrf1 regulates histone modifications and how the loss of Uhrf1 function leads to disruption of bivalent histone modifications. First, we investigated if modulation of Uhrf1 expression levels directly influenced histone modification. We either overexpressed or depleted Uhrf1 in mouse embryonic fibroblasts (MEFs) and mESCs, and then we examined H3K4me3, H3K27me3, and H3K9me3 modifications. We observed large changes in H3K9me3 and H3K4me3 that are strongly correlated with the expression levels of Uhrf1 (Supplementary Fig. 4a), suggesting that Uhrf1 is not only enriched on the bivalent loci, but also regulates global active histone modification levels.

Uhrf1 does not have a catalytic domain for HMT activity[28], indicating that this euchromatic function of Uhrf1 may be regulated via interplay with other catalytic enzymes. In order to identify proteins that interact with Uhrf1 and modulate histone modification, we analyzed the protein interactome of Uhrf1. To enable this, we established the J1 ESC lines that expressed sub-endogenous level of FLAG-tagged and biotinylated Uhrf1 (Supplementary Fig. 4b)[29]. Multi-protein complexes were purified using one-step anti-FLAG and streptavidin affinity capture

combined with mass spectrometry analysis. We identified 159 Uhrf1-interacting proteins, including Dnmt1, which is a well-known binding partner of Uhrf1[8] (Supplementary Data 1). GO analysis determined that many interacting proteins were related to cell cycle and chromatin structure regulation (Supplementary

Fig. 4c). Comparative analysis of our Uhrf1 interaction data against the public interactome databases identified that Uhrf1 interacts with PRC2, the Kap1 complex, and Dnmt1. Interestingly, the Setd1a/COMPASS complex proteins that catalyze the

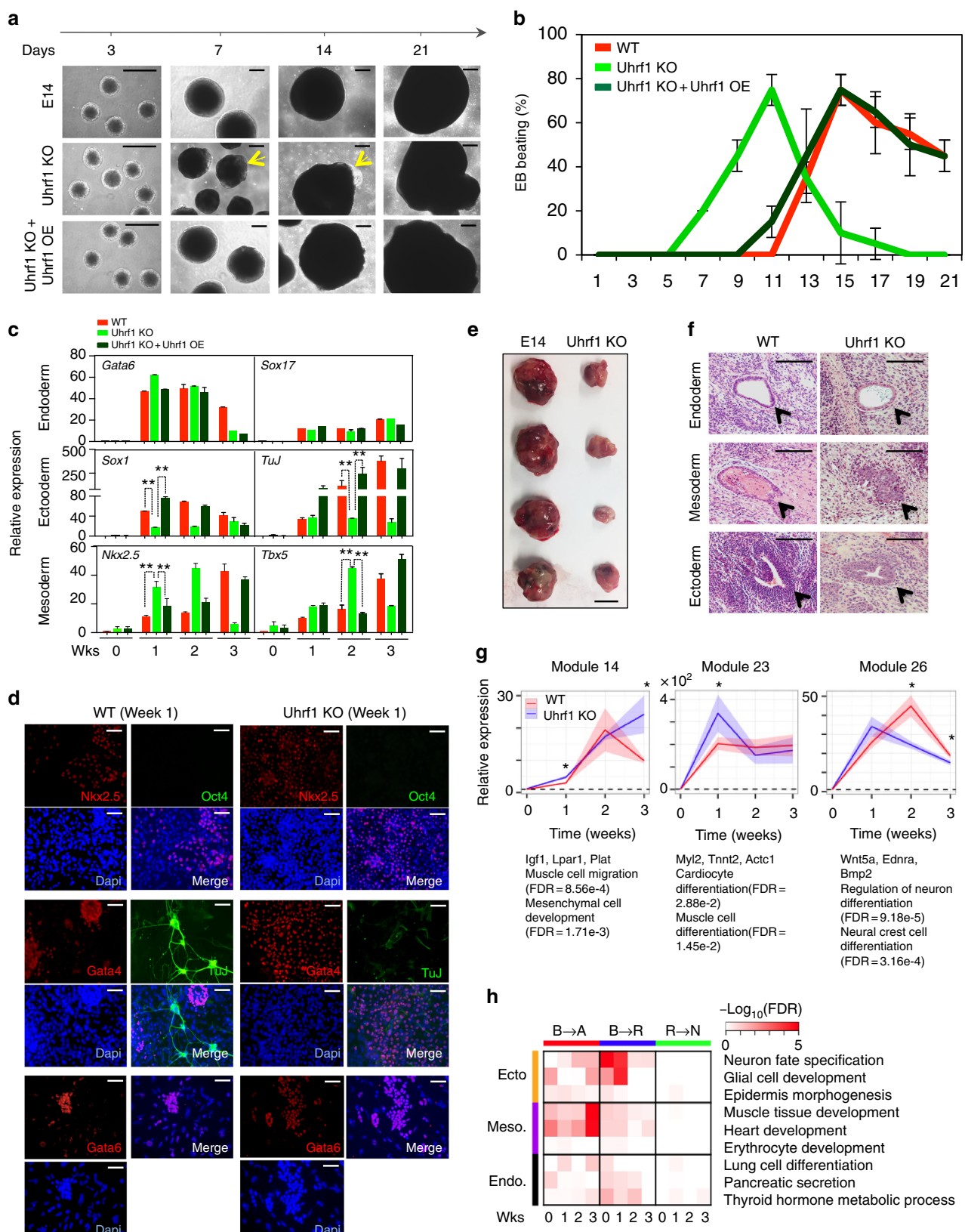

methylation of histone H3 at lysine 4 were found to interact with Uhrf1 (Supplementary Fig. 4d).

The way Uhrf1 recognizes and modifies histone H3 at lysine 9 is already well established[9–11]. H3K27me3 cooperates with H3K9me3 for heterochromatin formation by maintaining an abundance of heterochromatin protein 1 alpha (HP1a)[30]. It also has been reported that Dnmt1 mediates an interplay between DNA methylation and H3K27me3 at pericentric heterochromatin, and Uhrf1 KO ESCs can affect the kinetics of H3K27me3 modification through a Dnmt1-dependent DNA methylation mechanism[31]. However, to our knowledge, how Uhrf1 regulates H3K4me3 modification still remains elusive. Therefore, we focused on identifying the molecular mechanism of H3K4me3 regulation through Uhrf1.

First, we confirmed the interaction of Uhrf1 with Setd1a and its core complex proteins by co-immunoprecipitation (Co-IP) (Fig. 3a). Furthermore, to identify the specific domain of Uhrf1 that mediated the Setd1a interaction, a series of domain-specific mutants were generated, and assessed for binding with Co-IP. Notably, deletion of the SRA domain abolished the interaction of Uhrf1 with Setd1a (Fig. 3b). Interestingly the SRA domain alone was sufficient to interact with Setd1a (Supplementary Fig. 4e). In addition, the SET domain of Setd1a is critical for interacting with Uhrf1 (Fig. 3c). To clarify whether Uhrf1–Setd1a interaction is direct or mediated through other proteins, we performed in vitro interaction assays with recombinant proteins. We purified recombinant Uhrf1 and used this as bait for interaction assay. For Setd1a, we used the recombinant human partial SETD1A-198H proteins (Amino acid, 1418–end), which contains n-SET and SET domains and has an identical amino acid sequence to the murine Setd1a protein in SET domain (Supplementary Fig. 4f). When we performed in vitro binding assay with these two recombinant proteins, we clearly demonstrated their direct interaction (Supplementary Fig. 4g).

From these data, we postulated that Uhrf1's ability to regulate H3K4me3 was potentially through its interaction with and recruitment of the Setd1a/COMPASS complex. To test this, we performed an in vitro HMT assay[22]. Uhrf1 complex was immunoprecipitated from ESCs and used for subsequent HMT assays. The affinity-purified Uhrf1 complex demonstrated methyltransferase activity (Fig. 3d), and western blot analysis further revealed that the Uhrf1 complex-mediated methylation on lysine 4 of Histone 3 (Fig. 3d). To verify that Setd1a is the major protein mediating H3K4 methylation in Uhrf1 complex, we knocked down Setd1a using shRNA in ESCs and then performed the HMT assay. We found that the depletion of Setd1a dramatically reduced the H3K4 methylation activity of the Uhrf1

complex, demonstrating that Uhrf1 controls the H3K4 methylation through its interaction with the Setd1a complex (Fig. 3e).

To further understand the role of Uhrf1 domains in H3K4 methylation, we next tested a series of domain-deleted Uhrf1 for HMT activity toward H3 substrates. The deletion of the SRA domain in Uhrf1 abolished its tri-methylation activity of H3K4, while deletion of the PHD or the TTD domains had no affect on its HMT activity (Fig. 3f). As shown in the interaction assay, SRA domain alone that is necessary and sufficient to interact with Setd1a was sufficient to induce the HMT activity, as well as H3K4me3 modification (Supplementary Fig. 4h). Interestingly, when we used assembled chromatin substrates, which includes DNA and core histones (H3, H4, H2A, and H2B), the deletion of TTD and PHD affected the total HMT activity, although H3K4me3 was not abolished as assessed by western blotting (Supplementary Fig. 4i). This suggests that the PHD and TTD domains are necessary for Uhrf1's function on naive chromatin, including DNA methylation or histone modifications other than H3K4me3. Overall, our results demonstrate that the SRA domain alone is essential for mediating H3K4me3 modifications by a direct interaction with Setd1a.

**Uhrf1 is essential for reprogramming by recruiting Oct4.** Our data demonstrate that Uhrf1 plays important roles in lineage specification of ESCs via faithful maintenance of histone modifications. Interestingly, we observed a dramatic up-regulation of Uhrf1 and Setd1a/COMPASS complex genes with reprogramming factors (Fig. 4a and Supplementary Fig. 5a). The expression levels of Uhrf1 peaked at day 13, when reprogramming is completed. We thus hypothesized that Uhrf1 is critical in the acquisition of pluripotency. Notably, overexpression of Uhrf1 in MEFs dramatically increased the reprogramming efficiency with three reprogramming factors (Oct4, Sox2, Klf4) but not with four factors (Oct4, Sox2, Klf4, and Myc) (Fig. 4b). Pluripotency of induced pluripotent stem cells (iPSCs) derived from OSK+Uhrf1 (iOSKU-13) was validated by qPCR, FACS analysis, and IHC analysis against a battery of ESC markers (Fig. 4c, d and Supplementary Fig. 5b). Furthermore, teratoma assay and chimera formation supported that iOSKU-13 was indeed pluripotent (Fig. 4d). We also tested the role of Uhrf1 in reprogramming by deleting Uhrf1. MEFs were isolated from Uhrf1[fl/fl] embryos, and Uhrf1 was subsequently deleted by expression of Cre recombinase (Supplementary Fig. 5c). Within a week of deletion, Uhrf1 KO MEFs showed normal proliferation (Fig. 4e), but failed to undergo proper reprogramming (Fig. 4f). Depletion of Uhrf1 by shRNA further supported the necessity of Uhrf1 for somatic cell reprogramming (Supplementary Fig. 5d and Supplementary

**Fig. 2** Uhrf1 deficiency induces neural differentiation defects. **a** In vitro EB formation of WT, Uhrf1 KO, and Uhrf1 KO ESCs expressed with Uhrf1. ESCs were differentiated for 3 weeks as EBs. Arrows indicate the cystic hollow. The scale bar represents 4 mm. **b** Appearance of spontaneous contractile activity during EBs growth. EBs were examined daily under the microscope for the presence of a rhythmic beating until day 21. Data are from $n = 3$ independent experiments. **c** qPCR analysis of lineage- specific markers and pluripotency marker upon in vitro differentiation. Data are expressed relative to WT ESCs values and values are normalized to β-actin. Data are from $n = 3$ independent experiments. The data values are mean ± s.d. **P < 0.05 by unpaired, two-tailed Student's t-test analysis. **d** Immunostaining of differentiated EBs. EBs were treated with Trypsin–EDTA at EB week 1 and plated onto gelatin-coated plates. Immunostaining with lineage-specific markers was performed 2 days after plating. The scale bar represents 4 mm. **e** Teratomas derived from immune deficiency mice. Scale bar represent 1 cm. **f** Representative histological sections from teratomas that developed in immunodeficient mice following injection with WT and Uhrf1 KO ESCs. Haematoxylin and eosin (H&E) staining reveal characteristic tissues from the endoderm, mesoderm, and ectoderm. The data are represented as $n = 6$ experimental replicates. The Scale bar represents 4 mm. **g** Expression pattern of transcriptional modules related to muscle differentiation and neuronal differentiation. Thick red and blue lines represent average relative gene expression to WT at week 0 in WT and Uhrf1 KO, respectively. Ribbon colors represent SEM. Representative genes and overrepresented GO terms are shown. Asterisk (*) represents significant difference (>1.25 fold change and Student's t-test p-value < 0.05) of gene expression between WT and Uhrf1 KO in each module. **h** Disruption of bivalent and repressive promoters in developmental genes. Over representation of GO terms in genes with changes from bivalent to active (B–A) and to repressive (B–R) and from repressive to no mark (R–N) is shown by heat colors

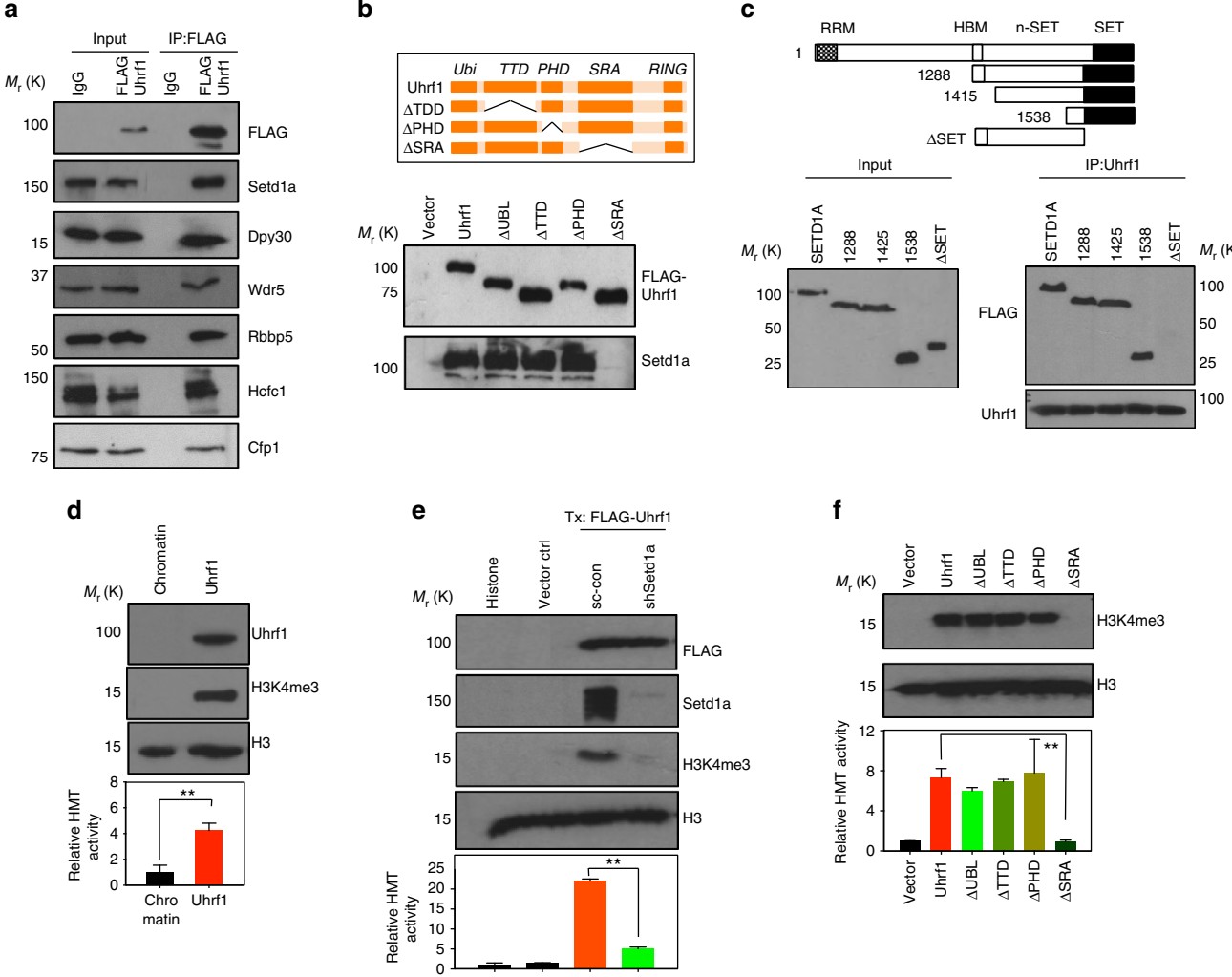

**Fig. 3** H3K4 methylation by Uhrf1 complex is mediated via a Setd1a. **a** Immunoprecipitation analysis of FLAG-tagged Uhrf1 protein. Uhrf1 was co-immunoprecipitated with Setd1a/COMPASS complex proteins isolated from mESCs. The data in c−g are represented as $n = 3$ experimental replicates (independent HMT assay from different cell cultures). Values are mean ± S.D. **$P < 0.01$, by unpaired, two-tailed Student's $t$-test analysis. **b** Mutants of Uhrf1 deleted of specific domain (ΔUbi, Ubiquitin domain; ΔTTD, TTD domain; ΔPHD, PHD domain; ΔSRA, SRA domain) were used for co-immunoprecipitation with Setd1a. **c** SET domain of SETD1A is necessary for interacting with Uhrf1. Series of SETD1A constructs with given deletions were used for Co-IP with Uhrf1. **d** Uhrf1 complex isolated form mESCs was used for assaying the histone H3K4me3 methyltransferase activity with chromatin as substrate. **e** Setd1a is a histone methyltransferase in complex with Uhrf1. Depletion of Setd1a reduced the H3K4 methyltransferase activity in Uhrf1 complex. **f** SRA domain is essential for a histone methyltransferase activity of Uhrf1 complex. In vitro HMT assay with each of Uhrf1 deletion mutant constructs was performed with H3 as substrate. H3K4me3 modifications were not detected with SRA domain deletion

Fig. 5e). These results imply that Uhrf1 potentially modulates pluripotent regulatory networks.

Oct4, Sox2, and Nanog are core transcriptional factors regulating genes associated with pluripotency[32]. Co-IP experiments revealed a significant interaction of Uhrf1 with Oct4 (Fig. 5a). In addition, Uhrf1 weakly interacted with Sox2, but not with Nanog. To determine the mechanism how Uhrf1 regulates pluripotency, we performed ChIP-seq against the core pluripotency transcription factors, Oct4, Sox2, and Nanog, both in WT and Uhrf1 KO ESCs. Despite a limited change in their expression, Oct4 binding was dramatically decreased in Uhrf1 KO ESCs, whereas Sox2 and Nanog binding was increased (Fig. 5b). Approximately half of WT-specific Oct4-binding sites were replaced by either Sox2 or Nanog in Uhrf1 KO cells, and the remainder showed a loss of Oct4 binding without any replacement(Fig. 5c). Specifically, genes involved in stem cell maintenance were enriched in regions showing replacement of loss of Oct4 binding with Sox2 and Nanog (Supplementary Fig. 6a, FDR

= 4.96e−4), suggesting that Sox2 and Nanog compensates for the dysregulation of Oct4 binding to maintain pluripotency in Uhrf1 KO cells.

To assess the effect of loss of Oct4 binding in Uhrf1 KO cells on chromatin structure, we performed assay for transposase accessible chromatin with high-throughput sequencing (ATAC-seq). Depletion of Uhrf1 significantly decreased chromatin accessibility in locations where Oct4 binding was lost, but not in locations replaced with either Sox2 or Nanog (Fig. 5d, and S6B, $p = 1.17e−10$). The loci showing loss of Oct4 binding upon Uhrf1 KO were correlated with regions highly enriched with Uhrf1 (Fig. 5e, $p < 2.2e−16$). We also observed that H3K4me3 was significantly reduced with loss of Oct4 binding (Fig. 5f, $p < 2.2e−16$), indicating that Uhrf1 directly regulates Oct4 binding by mediating an active open chromatin conformation.

Genes where Oct4 binding was lost but Sox2 or Nanog replacement was not made included mesodermal, neuroectodermal, and pluripotent genes (Supplementary Fig. 6c). Thus, the

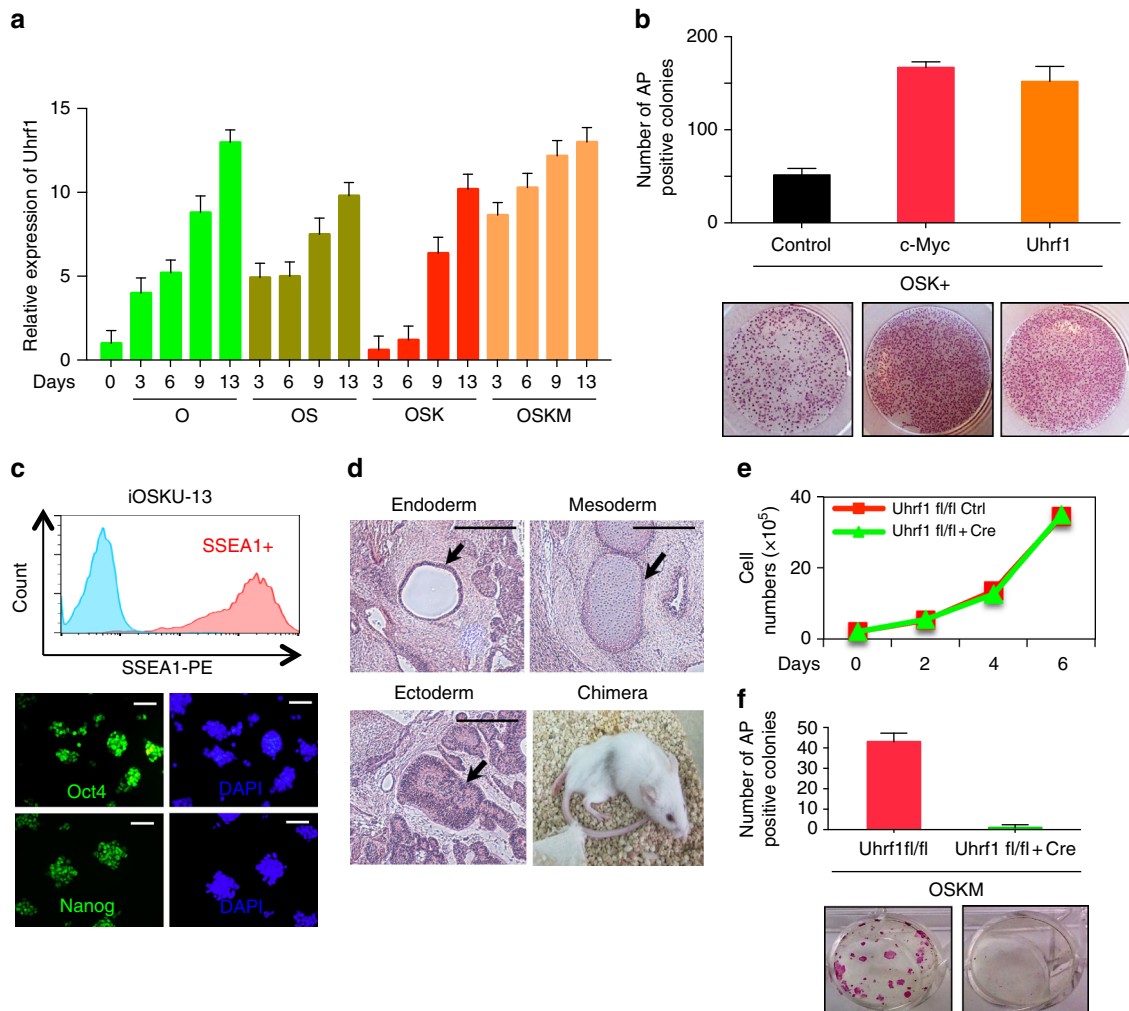

**Fig. 4** Uhrf1 increases the reprograming efficiency. **a** qRT-PCR analysis of *Uhrf1* during reprogramming. Error bars represent mean ± s.d. **b** Efficiency of iPSC colony formation was assessed by alkaline phosphatase (AP) staining after reprogramming. These are representatives of at least *n* = 3 experimental replicates. **c** Histogram of FACS flow of SSEA-1 and immunostaining images of Oct4, and Nanog of the iOSKU-13 obtained by reprogramming MEFs with Oct4, Sox2, Klf4, and Uhrf1. DAPI was used for counter-staining. The scale bar represents 4 mm. **d** Teratoma analysis showing tissues of three germ layers (respiratory epithelium for endoderm, cartilage for mesoderm, and neural tissue for ectoderm) and chimera generated from iOSKU-13 iPSC line. The scale bar represents 4 mm. **e**, **f** MEFs isolated from Uhrf1$^{fl/fl}$ mice were treated with Cre-retrovirus. Uhrf1 KO MEF cells were used for cell proliferation assay (**e**) and reprogramming (**f**). The number of AP + iPSC colonies was analyzed. The data in e–f represent as *n* = 3 experimental replicates

loss of Oct4 binding potentially influences both development and pluripotency. The *Zscan4* gene cluster, which is specifically activated in 2-cell (2C) embryos and essential for iPSC formation, was one of the identified targets of the loss of Oct4 binding (Fig. 5g)[33]. *Zscan4* family was the most down-regulated genes in Uhrf1KO ESCs (Fig. 5h). We also identified previously uncharacterized genes (*Gm8300, BB287469, Gm2022, Gm5662, Gm21312, Gm21293,* and *Gm21304*) that are activated in the 2C stage and are included in top 20 down-regulated genes in Uhrf1KO ESCs (Fig. 5h and Supplementary Figs. 6d). Similarly with *Zscan4*, these 2C genes form gene clusters in specific chromosomal loci (*Gm8300-2022* and *Gm21312-21304*) (Fig. 5h and Supplementary Fig. 6d). The loss of Oct4 binding was also observed in the vicinity of these 2C clusters along with active chromatin conformation changes (Fig. 5g and Supplementary Fig. 6e, f). Furthermore, a 2C-specific retrotransposon, *MERVL,* was also significantly down-regulated with the loss of Oct4 binding (Supplementary Fig. 6g, h). Therefore, 2C genes appear to be a direct target of Uhrf1 and Uhrf1-mediated Oct4 binding. Overall, our results indicate that Uhrf1 regulates both embryonic cell development and somatic cell reprogramming by mediating

chromatin conformation changes and the binding of core pluripotent factors.

## Discussion

Previous studies have shown that Uhrf1 is critical in maintaining DNA methylation by regulating Dnmt1 and forming heterochromatin as a reader of H3K9me3[4,24]. In this study, we have identified a novel function of Uhrf1 in ESCs, in the maintenance of bivalent chromatin structures through the interaction with the Setd1a/COMPASS complex protein.

Although a number of phenotypes in Uhrf1 KO cells (i.e. cell death and small teratoma) overlapped with the Dnmt1 KO[34], importantly we observed a number of unique features in this Uhrf1 KO study. Transcriptome analysis of Dnmt1 and Uhrf1 KO ESCs demonstrated distinct profiles that segregated the two KO phenotypes with regards to pluripotency and developmental gene expression. Global histone modification analyses by ChIP-seq also revealed that the deletion of Uhrf1 induced a significant disruption in both H3K4me3 and H3K27me3 histone modifications, mainly on bivalent promoters (Fig. 1). The observed

bivalent disruption in Uhrf1 KO implicates Uhrf1 in the transcriptional dysregulation of lineage specification markers during differentiation (Fig. 2). This was further supported during spontaneous EB differentiation of Uhrf1 KO ESCs, which showed a higher expression of cardiac developmental genes, but reduced expression of neuroectodermal genes (Fig. 2). Notably, cardiac

developmental genes displayed loss of H3K27me3, whereas significant reduction of H3K4me3 was detected on neuroectodermal genes.

We demonstrated that the interaction of Uhrf1 with the Setd1a complex proteins is critical for maintaining promoter bivalence and neuroectodermal specification in ESCs. Like Uhrf1 KO ESCs,

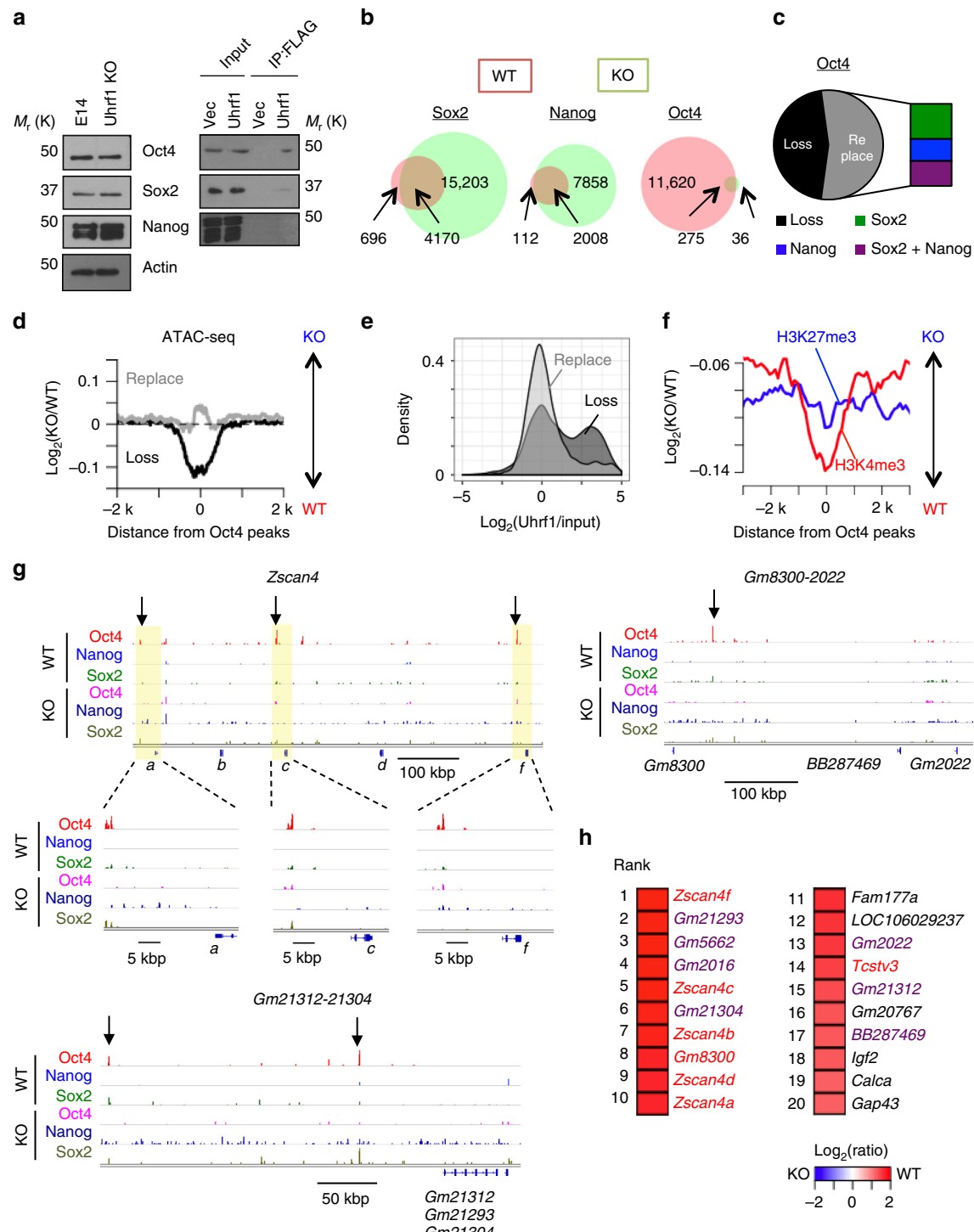

**Fig. 5** Uhrf1 modulates core pluripotent factor binding. **a** Co-IP experiment of Uhrf1 with core pluripotent factors. **b** Venn diagrams showing differential binding of core pluripotent factors between WT and Uhrf1 KO ESCs. **c** Ratio of loss and replacement of Oct4 binding. **d** ATAC-seq read distribution around Oct4-binding sites. $\log_2$ (KO/WT) of normalized ATAC-seq read count was drawn in the location of loss (black) and replacement (gray) of Oct4 binding, separately. **e** Enrichment of Uhrf1 binding on Oct4-binding sites. **f** ChIP-seq read distribution for H3K4me3 and H3K27me3 around the location of the loss of Oct4 binding. $\log_2$ (KO/WT) was drawn. **g** Binding landscape of core pluripotent factors in 2C-specific gene clusters. **h** Top 20 down-regulated genes in Uhrf1KO ESCs. Known and previously uncharacterized 2C genes were shown by red and purple colors[33]

Setd1a KO ESCs also present with defects in neuroectodermal differentiation[20]. Interestingly, Mll2 KO ESCs showed delayed cardiomyocyte differentiation rather than neuroectodermal defect, further supporting that Uhrf1 regulates neuroectodermal specification with the Setd1a/COMPASS complex[35]. As stated, neuroectodermal genes are down-regulated in Uhrf1 KO EBs, but are upregulated in Dnmt1 KO EBs[36]. Another key interesting difference is that Dnmt1 KO cells successfully undergo differentiation to mature astrocytes[37]. Our data demonstrated that the disruptions of bivalent marks in Uhrf1 KO are not due to disruptions in H3K9me3 or Dnmt1, but partly the loss of Setd1a function, which results in the reduction of H3K4me3 modification on neuroectodermal gene promoters.

In contrast to H3K4me3-mediated neuroectodermal lineage regulation, Uhrf1 KO cells showed a reduction of H3K27me3 on mesodermal lineage genes. One potential explanation is that the decrease of H3K27me3 in Uhrf1 KO may be due to the failure to recruit PRC2 at mesodermal gene promoters. This is further supported by our proteomics study showing the direct interaction of Uhrf1 with Rbbp4, which is one of components of PRC2. In addition, PRC2-deficient ESCs increased expression of mesodermal markers[38]. Because of the accompaniment and crosstalk of repressive epigenetic modifications[30,31], Uhrf1-mediated PRC2 recruitment may be more linked with H3K9me3 and Dnmt1. As observed in Uhrf1 KO EBs, Dnmt1 KO EBs activate mesodermal genes earlier than WT[39]. The detailed molecular mechanisms underlying skewed mesodermal lineage differentiation in Uhrf1 KO cells still require further investigation.

Regulation of euchromatic genes by Uhrf1 is poorly understood. The Shi lab has demonstrated that Uhrf1 recognized histone H3 unmodified arginine 2 (H3R2) with its PHD domain and is involved in activating target genes[11]. In our study we found that Uhrf1 interacts with Setd1a/COMPASS complex proteins and directs the catalytic activity of Setd1a for H3K4 methylation. In addition, the PHD and TTD domains of Uhrf1 are dispensable for binding Setd1a (Fig. 3b and Supplementary Fig. 4e) and also methylation of H3 substrate (Fig. 3f). These domains nevertheless seem essential for the interaction and methylation of naive chromatin (Supplementary Fig. 4i). It is known that H3R2 methylation antagonizes H3K4 methylation[40,41]. Thus, binding of unmodified H3R2 by Uhrf1 could potentially mediate or prime catalytic modification of H3K4me3 by Setd1a/COMPASS complex.

The importance of iPSCs in the context of cell therapeutics and disease modeling has led to consolidated efforts to understand the molecular basis of reprogramming, including epigenetic regulators for DNA methylation and histone modification[42,43]. Dnmt3a was shown to be dispensable for reprogramming[44], but inhibition of Dnmt1 accelerates reprogramming[45]. Proteins of the PcG and TrxG complexes have been tested for their roles in reprogramming[46]. Setd1a and Wdr5 are essential for reprogramming, while Ezh2, Suz12 and Eed were shown to be dispensable[47,48]. In this study, we observed a ~100-fold increase in reprogramming efficiency of MEFs when Uhrf1 was included with a three-factor cocktail of Oct4, Sox2, and Klf4 (Fig. 4). Because members of the Setd1a/COMPASS complex interact with Uhrf1 (Fig.3a), Uhrf1 may recruit the Setd1a/COMPASS complex proteins to reprogramming targets, and facilitate the formation of pluripotency transcriptional networks. Indeed, overexpression of Uhrf1 in MEFs increased H3K4me3 level, while loss of Uhrf1 in ESCs interrupted binding of the core pluripotency factors to active chromatin regions. In addition, Uhrf1 KO disrupted Oct4 binding at 2C-specific genes with disruption of active chromatin conformation and a concomitant reduction of expression. Since Setd1a was previously proposed as Oct4 co-activator in pluripotent stem cells, the mediation of H3K4me3 by Uhrf1 may be

an important step for 2C-gene activation[48,49]. The down regulation of 2C genes could be due to reduced numbers of 2C-like cells. The 2C-like state is transient, with a non-2C state observed under conventional ESC medium[49], proposing the existence of feedback or feed forward regulation. Previously, Zscan4 has been reported to degrade Uhrf1 for telomere maintenance in ESCs[50]. Therefore, the interdependent control of Uhrf1 and Zscan4 is one of potential mechanisms for the interconversion between 2C-like and non-2C state. Overall, our work identified Uhrf1 as a key regulator of differentiation and pluripotency through regulation of Setd1a/COMPASS-mediated histone modification.

Collectively, the findings presented in this study demonstrated that Uhrf1 has an essential function in regulating bivalent promoters during reprogramming, pluripotency and differentiation. Additionally, our studies defined a molecular mechanism for Uhrf1-mediated bivalent chromatin regulation, in which Uhrf1 directly interacts with the Setd1a/COMPASS complex to mediate the deposition of H3K4me3 modifications for pluripotency.

## Methods

**Cell culture**. E14 and Uhrf1 KO ES cells were obtained from the Koseki group[4] and maintained in mES medium (DMEM supplemented with 15% FBS, 1% non-essential amino acids, 2 mM glutamine, 100 U/mL penicillin/streptomycin, 0.1 mM β-mercaptoethanol, and 1000 U/mL ESGRO leukemia inhibitory factor) on 0.1% gelatin or MEF-coated plates, incubated at 37 °C and passaged every second or third day. Uhrf1^{fl/fl} mice were purchased from EMMA (B6N-Uhrf1^{tm1a(EUCOMM)Wtsi}/Ieg). Animals were handled according to protocols approved by the Yale University Animal Care and Use Community. ESCs were isolated from E3.5 embryos and MEFs were prepared from E13.5 embryos and were cultured in mES medium and 10% FBS supplied DMEM, respectively. Retrovirus expressing Cre recombinase from pBABEpuro-Cre vector (#1764, Addgene) was used to generate the Uhrf1 KO cell lines used in this study.

**Generation of Uhrf1 transgenic cell lines**. To generate mouse BirA-ESC lines that stably expressed both FLAG and biotin-tagged Uhrf1, BirA-ESCs were transfected with Uhrf1 constructs using the Amaxa nucleofector (Lonza) according to the manufacturer's protocol. Transfected cells were cultured for 10 days in growth medium supplemented with puromycin. Drug-resistant clones were subsequently selected based on Uhrf1 expression.

**Reprogramming**. Retroviruses containing the reprogramming factors (Oct4, Sox2, Klf4, and Myc) were generated using HEK293T cells[27]. Briefly $1 \times 10^5$ cells were plated and transfected the next day with reprograming factors using X-tremeGENE 9 according to manufacturer's instructions. Two days after transfection, supernatants were collected and concentrated via ultracentrifuge. The titer of each viral supernatant was calculated. The virus mixture (MOI = 5 for each factor) was then applied to $1 \times 10^5$ MEFs. One day after virus infection, medium was replaced with DMEM medium supplemented with 10% FBS. After 3 days, cells were replated onto plates pre-seeded with irradiated MEFs. The cells were cultured with mESCs medium throughout the reprogramming. Successfully reprogramed cells were fixed with 4% formaldehyde in PBS and subsequently stained with the Alkaline Phosphatase Kit (Sigma-Aldrich) for analysis of reprogramming efficiency.

**In vitro HMT assay**. Wild type or mutant Uhrf1 protein complexes were purified using Uhrf1 antibody or FLAG antibody from mESCs. When preparing chromatin as substrate, HeLa core histones were assembled into chromatin using the chromatin assembly kit according to the manufactures instructions (Active motif). For each HMT assay reaction, 400 ng of assembled chromatin or histone H3 peptide were incubated with purified Uhrf1 complex in 20 µl reaction buffer (50 mM Tris–HCl, pH 8.5, 50 mM KCl, 5 mM MgCl₂, 0.1 mM EDTA and 10% glycerol) supplemented with 100 µM cold (for immunoblotting) or 1 µCi ³H-labeled (for fluorography) S-adenosylmethionine (SAM). Proteins were resolved by SDS-PAGE and subjected to immunoblotting (cold SAM) or scintillation recording (³H-labeled SAM). Relative HMT activity was calculated against IgG-enriched or vector-enriched lysate.

**In vitro Uhrf1 and SETD1A-binding assay**. GST-tagged recombinant human partial SETD1A protein (SETD1A-198H, amino acid 1418–end) was purchased from Creative BioMart Company (Cat #SETD1A-198H). In order to purify recombinant Uhrf1, Uhrf1 was cloned into pGEX6p1, and transformed into BL21 (DE3) E. coli for purification[51]. Purified recombinant GST-Uhrf1 protein was incubated with SETD1A-1418 protein (0.15 and 0.3 µg) in binding buffer (25 mM HEPES, 100 mM NaCl, 0.02% Triton X-100, 5% Glycerol, 5% BSA, and 1 mM DTT). Uhrf1 was pulled down using an Uhrf1 antibody and agarose beads. The

bound proteins were separated on SDS-PAGE gel and used for the western blotting against a GST antibody

**RT-qPCR.** Total RNA was isolated using an RNeasy mini kit (Qiagen), and 1 µg of RNA was transcribed into cDNA using the iScript cDNA Synthesis Kit (Bio-Rad). RT-qPCR was performed in triplicate using iQ SYBR Green Supermix (Bio-Rad). Delta values were normalized with β-actin. Error bars represent mean ± SD of technical triplicates. All primers are shown in Supplementary Data 2.

**Western blotting.** Protein extracts for the detection of histone modifications were obtained by incubating cells of interest in RIPA buffer (25 mM Tris–HCl, pH8.0, 150 mM NaCl, 0.1% SDS, 0.5% sodium deoxycholate, 1% NP-40, and proteinase inhibitor cocktail), followed by sonication. Total protein was separated on a 15% PAGE gel. For the immunoprecipitation, cells were lysed with GST–IP buffer (50 mM Tri–HCl pH 8.0, 150 mM NaCl, 100 mM KCl, 1%NP-40, 2 mM EDTA, and proteinase inhibitor cocktail) for 15 min on ice and used for immunoprecipitation. The information of primary antibodies and dilutions is described in Supplementary Data 3. Uncropped scans of immunoblots are presented in Supplementary Figures 7, 8. The corresponding figure number and the molecular weight in kDa are indicated. Dotted boxes highlight the cropped areas presented in the figures.

**Immunostaining.** Cells were fixed with 4% formaldehyde/PBS fixation solution for 15 min and then washed with PBS three times. After permeablization with 0.1% Triton X-100/PBS solution, cells were subjected to blocking and then incubated with primary antibody in a 3% BSA/PBS solution. Alexafluor 488 and 555 were used for fluorescence detection, and DAPI was used for a counter-staining. All antibodies used in this study were listed in Supplementary Data 3.

**Chromatin immunoprecipitation assay.** Approximately $2 \times 10^7$ ESCs cells were cross-linked by the addition of formaldehyde (1% final concentration) for 15 min on a rotator. The cross-linking process was stopped with 0.125 M glycine for 10 min, followed by three washes in PBS. The cells were resuspended in isotonic buffer supplemented with 1% NP-40 to isolate the nuclei. The isolated nuclei were then resuspended in ChIP buffer (20 mM Tris–HCl pH 8.0, 150 mM NaCl, 2 mM EDTA, and protease inhibitors). Extracts were sonicated using for three runs of 10 cycles of 15 s "ON", 15 s "OFF" at a high power setting. Cell lysate were centrifuged at 12,000×g for 10 min at 4 °C. The supernatant was diluted with ChIP dilution buffer (20 mM Tris–HCl pH 8.0, 150 mM NaCl, 2 mM EDTA, 1% Triton) before the immunoprecipitation step. H3K4me3 (07-473, Millipore), H3K27me3 (07-499, Millipore), H3K9me3 (ab8898, abcam), Oct4 (#5677, Cell Signaling Technology), Sox2 (#23064, Cell Signaling Technology), Nanog (#8785, Cell Signaling Technology), or Uhrf1 (sc98817, Santa Cruz) antibodies were then incubated overnight at 4 °C on a rotator. All antibodies used in this study are listed in Supplementary Data 3. Immunoprecipitated complexes were successively washed with washing buffer I (2% SDS), washing buffer II (50 mM HEPES pH 7.5, 500 mM NaCl, 0.1% deoxycholate, 1% Triton X-100, 1 mM EDTA), washing buffer III (10 mM Tris–HCl pH 8.1, 250 mM LiCl, 0.5% deoxycholate, 1 mM EDTA), and TE buffer (10 mM Tris–HCl pH 7.5, 1 mM EDTA). All washes were performed at room temperature for 8 min on a rotator. SDS elution buffer (50 mM Tris–HCl pH 8.0, 10 mM EDTA, 1% SDS) was added and incubated at 65 °C overnight in order to reverse crosslink protein–DNA complexes. After reversing the cross-linking, DNA was purified using a Purification Kit (Invitrogen) according to the manufacturer's instructions. Purified DNA was used for sequencing using an Illumina Genome HiSeq2000, or for qPCR using primers listed in Supplementary Data 2.

**Mass spectrometry and data analysis.** One-step affinity purification with SA or M2 beads was performed as described[52]. LC–MS/MS sequencing and peptide identification were performed by Taplin Biological Mass spectrometry Facility at Harvard Medical School with five biological replicates. Proteins with less than two unique peptide sequences and common background proteins were removed from subsequent analysis. The background protein sets were used from a previously reported list[52]. The number of unique peptides in each protein was normalized to the total number of unique peptides from all sequenced proteins. To identify candidate interaction partners, we searched our data for proteins that had a two-fold higher normalized count of unique peptides in our biotinylated Uhrf1 SA or FLAG pull-down libraries when compared to a control BirA-SA or FLAG pull-down library in at least one replicate of our Uhrf1 pull-downs.

**In vitro differentiation.** EB differentiation was initiated by forming hanging drops. Cells were suspended with EB differentiation medium (IMDM containing with 15% FBS, 200 µg/ml transferrin, 0.05 ng/ml ascorbic acid, 1 mM sodium pyruvate, 100 U/mL penicillin/streptomycin, 2 mM glutamine, and 450 µM mono-thiolglycerol). Multiple 200 cells/30 µL aliquots were pipetted onto bacterial Petri dishes that were inverted for 3 days at 37 °C, after which the EBs were flushed with EB differentiation medium and subsequently cultured as 3D spheroids.

**mRNA library preparation for next generation sequencing.** mRNA was isolated from 5 µg total RNA using Dynabeads mRNA DIRECT (Invitrogen) and was fragmented with RNA fragmentation reagent (Ambion). First strand cDNA synthesis was done using the SuperScript III First-Strand Synthesis System and 3 µg µl$^{-1}$ random hexamers (Invitrogen), followed by second strand synthesis with DNA Polymerase I and RNase H. After purification, a sequencing library was generated from the double-stranded cDNA by using paired-end adaptors (Illumina) and the NEBNext DNA Sample Prep Reagent Set 1 (NEB). This library was sequenced using an Illumina Genome HiSeq2000.

**ATAC-seq.** 50,000 cells were lysed in cold lysis buffer (10 mM Tris–HCl, pH7.4, 10 mM NaCl, 3 mM MgCl₂, 0.1% IGEPAL CA-630)[53]. After spinning at 500×g for 5 min, the pellet was used for transposition reaction by Nextera® DNA Library Prep Kit. Following purification by Qiagen EinElute kit, ATAC-seq library was constructed and amplified by NEB High Fidelity 2xPCR master mix. Finally, the library was purified by AxyPrepTM Mag PCR Clean-up kit.

**Data processing and analysis of RNA-seq, ChIP-seq, and ATAC-seq.** Mouse genome sequences (version mm9) and the genomic coordinates of RefSeq genes and the repetitive element RepeatMasker were downloaded from UCSC genome browser. All RNA-seq reads were mapped into mm9 mouse genome using Tophat (v1.1.4), Samtools (v0.1.18), and Bowtie (v0.12.7) with default parameters[54]. Normalized expression value (reads per kilobase per million mapped reads (RPKM)) was calculated by Cufflinks (v1.1.0) using RefSeq genes as reference annotation by "--GTF" option[55]. Public transcriptome data from Dnmt1 KO ESCs (SRX695151 and SRX695157) and preimplantation embryos (SRP055882) was downloaded from the NCBI short read archive (SRA)[26,56]. Genes with more than a 1.5-fold change during at least one time point were used to identify transcriptional modules. Euclidean distances were calculated from Log₂-transformed RPKM values between all gene pairs with the dist function and were subsequently used for Ward. D2 hierarchical clustering via the hclust function in R. Transcriptional modules were then identified by a 20 height cutoff. GOstats in the Bioconductor package was used to evaluate the overrepresentation of GO terms. The Benjamini and Hochberg (BH) method was used for multiple test correction by the p.adjust function. The enrichment of early developmental genes and stem cell functions was analyzed by GSEA (v2.1.0) software[57]. Log₂-transformed RPKM values were used for GSEA with 100 permutations of gene sets, classic enrichment statistic and signal-to-noise metric. Gene sets for early development and stem cell functions were obtained from a previous publication[27].

ChIP-seq and ATAC-seq reads were mapped to mm9 genome by Bowtie2 (v2.1.0) with options "--local -D 15 -R 3 -N 1 -L 20 -i S,1,0.50 -k 1". Read count was normalized as total number of mapped reads. Uhrf1 enriched and depleted regions were assigned when there was more than two-fold difference between ChIP and input and less than a 0.05 adjusted p-value within 10k-bp sliding window. The p-value was estimated by Poisson distribution from the ChIP-seq read counts compared to the normalized input counts[58]. For histone modifications, peak regions were identified by the rseg-diff program in RSEG software (v0.4.8) with parameters "-i 20 −v −mode 2" and the 50bp-deadzone correction file[59]. Non-overlapped peak regions between WT and Uhrf1KO were defined as differentially modified regions. Promoter configuration was classified by H3K4me3 ChIP-seq reads within ±2kbp of the TSS and by H3K27me3 within ±5kbp. Combination of log2(H3K4me3/input) > 1 & FDR<0.05 by Poisson test, and log2(H3K27me3/input) > 1 & FDR<0.05 by Poisson test were used for classification, as follows. Active promoter: presence of H3K4me3 and absence of H3K27me3; bivalent: presence of H3K4me3 and presence of H3K27me3; repressive: absence of H3K4me3 and presence of H3K27me3; no mark: absence of H3K4me3 and absence of H3K27me3. The retention rate of promoter types was calculated by comparing KO to WT at each time point.

Oct4, Nanog, and Sox2 binding sites were identified by MACS peak caller (v1.4.2) with "-g mm" option. Loss and gain of their binding by Uhrf1 depletion was evaluated by comparing MACS peaks between WT and Uhrf1-/-. Peaks within 100-bp distance were considered as overlapping. Furthermore, WT-specific Oct4 peaks were compared with KO-specific Sox2 and Nanog peaks. We defined WT-specific Oct4 peaks overlapped with and without KO-specific Sox2 or Nanog peaks as "replacement" and "loss", respectively. For example, "Sox2 + Nanog replacement" means that Oct4 alone peak in WT cells was changed into Sox2 and Nanog peak in KO cells. The closet genes to Oct4 peaks were used as target genes.

ATAC-seq reads, whose fragment size is lower than mono-nucleosome size (=150 bp), were used for subsequent analyses. Genomic coverage of ATAC-seq reads was calculated by genomecov function in Bedtools (v2.25.0). Read count was normalized by total number of mapped reads. ATAC-seq reads within Oct4 peaks were also counted. The difference of chromatin organization and histone modifications between WT and Uhrf1-/- ESCs was evaluated by comparing the normalized read count of ATAC-seq and ChIP-seq within Oct4 peaks with T test.

All high-throughput sequencing data produced by this study were listed in Supplementary Data 4. (h) MeDIP-seq (ERP000570) in E14 ESCs were obtained from SRA[25]. Reads were mapped to mouse genome by Bowtie2 as described above.

**Chimera formation**. iPSC line (iOSKU-13) derived from MEF cells (B6/DBA2 mouse) by expressing Oct4, Sox2, Klf4, and Uhrf1 were injected into fertilized CD-1 blastocyst to identify the coat-color chimerism. Injection was performed by Yale Genome Editing Center. In brief, super-ovulated CD1 female was mated with CD1 male, and fertilized embryos were isolated. CD1 blastocysts were injected with iOSKU-13, and transferred to uterus of a foster female. Newborn mice were analyzed for coat-color chimerism.

**Statistical analysis**. Data were represented as means ± standard errors. Statistical significance was determined using Student's $t$-test. $p < 0.05$ was defined as statistically significant.

**Data availability**. All data generated or analyzed during this study are included in this article or the Supplementary Information files. RNA-seq, ChIP-seq, and ATAG-seq (GSE113915) datasets generated in this study have been deposited in the Gene Expression Omnibus (GEO).

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

## Acknowledgements

We thank all the Park lab members for their helpful comments and active discussion. We appreciate Dr. Haruhiko Koseki for sharing Uhrf1 KO ESC, and Dr. Nils Neurenkirchen and Dr. Haifan Lin for advice in purifying recombinant Uhrf1. J.W. was partly supported by NIH (R21HD087722). I.H.P. was supported in part by the NIH (GM0099130-01, GM111667-01), the CSCRF (12-SCB-YALE-11, 13-SCB-YALE-06, 16-RMB-Yale-04), the KRIBB/KRCF (NAP-09-3), and a CTSA grant UL1 RR025750 from the National Center for Advancing Translational Science (NCATS), a component of the NIH, and the NIH Roadmap for Medical Research. The contents are solely the responsibility of the authors and do not necessarily represent the official view of the NIH. The sequencing services performed for this study were conducted at Yale Stem Cell Center Genomics Core facility, which was supported by the Connecticut Regenerative Medicine Research Fund and the Li Ka Shing Foundation. Computation time was provided by the Yale University Biomedical High Performance Computing Center.

## Author contributions

K.-Y.K. and I.-H.P. conceived the study; K.-Y.K., Y.T., J.S., B.C., Y.X., B.P., J.D., Y.-W.J., J.-H.K., E.H., H.L., R.D., M.Z., J.-H.L., J.M.L., and D.S. performed experiments; Y.T., J. W., J.K., and I.-H.P. supervised the research; Y.T., J.W., and G.J.S. edited the manuscript; K.-Y.K., Y.T., and I.-H.P. wrote the manuscript.

## Additional information

**Competing interests:** The authors declare no competing interests.

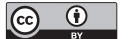

