## [Peer Review File · Nature Communications]

Reviewers' comments:

Reviewer #1 (Remarks to the Author):

Uhrf1 plays critical roles in maintaining DNA methylation and depositing histone H3K9me3 by recruiting Dnmt1 and KAP1/Setdb1, respectively. In this study, Kim et al. showed that Uhrf1 interacts with the Setd1/COMPASS complex and is essential for neuroectodermal specification and somatic cell reprogramming. These findings are potentially significant. However, the proposed mechanisms are not entirely supported by convincing experimental data.

Major issues:

1. It has been shown that Uhrf1 binding to hemimethylated DNA can inhibit Setdb1-mediated H3K9me3 deposition at some loci (Sharif et al., Cell Stem Cell 2016). Given that H3K4me3 and H3K9me3 are often antagonistic, it is possible that the decrease in H3K4me3 in Uhrf1-deficient cells is, at least in part, a secondary effect of H3K9me3 increases. The authors ought to conduct experiments to exclude that possibility or determine the relative significance of the Setd1/COMPASS and KAP1/Setdb1 complexes.
2. Some results are inconsistent. For example, the effect of FLAG-Uhrf1 is ~2000 fold over vector control in Fig. 3D, whereas it is only ~2.5 fold in Fig. 3E (sc-con vs. Vector con). Is it because of FLAG-Uhrf1 expression levels, cell type difference (HEK293 vs ES), or experimental variations? The blot and the bar graph in Fig. 3E are also inconsistent. Specifically, both the Vector control and sc-con samples show a faint signal in the H3K4me3 blot and do not seem to be different.
3. The authors claimed that the SRA domain mediates Uhrf1 interaction with Setd1a. Several issues need to be clarified: a) Is the interaction direct? b) Biochemical and structural data indicate that SRA domain binds DNA. Is Uhrf1-Setd1a interaction dependent on the presence of DNA? c) From Fig. S3C, deletion of UBL or TTD also seems to affect (albeit not abolish) the interaction. d) The methylation data in Fig. 3F are inconsistent with the mapping data in Fig. S3C.
4. The authors showed that Uhrf1^{-/-} MEFs failed to be reprogrammed by Yamanaka factors (Fig. 4E). Do these cells show defects in survival and proliferation compared to WT MEFs?

Minor issues:

1. A Uhrf1 blot is needed to show overexpression and the knockdown efficiency in Fig. S3A.
2. Page 5, lines 148-151. The authors described that more than half of repressive marks and bivalent marks in WT ESCs were lost in Uhrf1^{-/-} ESCs. Yet, their total fractions showed no obvious changes (Fig. 1D). Does that mean Uhrf1^{-/-} ESCs also gain repressive marks and bivalent marks in other genomic regions? Please clarify.
3. Fig. 2F. It would be helpful to define B-A, B-R, R-N in the figure legend.
4. Fig. 5G. It would be helpful to add scale bars (in kb) to indicate the relative locations of binding peaks and genes.
5. TTD (tandem Tudor domain) was misspelled as TDD throughout the manuscript.
6. Fig. 3F. shSetd1a was misspelled as shSeta1a. In Fig. S3F, shSet1A should be changed to shSetd1a to be consistent.
7. Fig. S2B. The x-axis should be indicated (weeks).
8. Figure S1H cited on Page 5, line 167 was not found in Figure S1.
9. Fig. S2C, lower panel. The labeling (H3K4me3, etc.) and the data were not aligned properly.

Reviewer #2 (Remarks to the Author):

Uhrf1 plays an important role in the inheritance of DNA methylation patterns during DNA replication. Despite the close relationship between Uhrf1 and accurate maintenance of these patterns, little is known about how Uhrf1 may influence pluripotency-specific epigenetic states. In this manuscript, Kim et al., suggest Uhrf1 helps maintain pluripotency and stabilize the mouse ESC epigenome through modulating active histone marks at bivalent loci. They further suggest this is through an interaction with the Setd1a/COMPASS complex to promote H3K4me3 and the

neuroectodermal lineage such that in the absence of Uhrf1, neuroectodermal lineage fate-restriction is limited. Additionally, Kim et al., show Uhrf1 is necessary for somatic cell reprogramming, particularly through recruitment of Oct4 to binding sites.

The main conclusion of this manuscript, that Uhrf1 regulates the neuroectodermal lineage through an interaction with the Setd1a complex to catalyze H3K4me3, is not convincingly supported. Key experiments supporting this model are hard to interpret (detailed below). Furthermore, despite the focus on the neuroectodermal lineage, Figures 1-3 show conversely (and perhaps more strongly) that Uhrf1 is important for repressing the mesodermal lineage and promoting H3K27me3. (Figure 1C middle panel, Figure S1A Plot 1 vs Plot 2, Figure S1B, Figure 2B-E, etc.), suggesting that promoting neuroectodermal fate-restriction is not the key mode of action of Uhrf1 in determining mESC pluripotency.

Figures 4 and 5 present highly interesting and suggestive data concerning Uhrf1's role in somatic cell reprogramming; this data could be very important for our understanding of molecular mechanisms regulating iPSC reprogramming. The evidence showing Uhrf1 is almost as efficient at promoting MEF reprogramming as c-Myc is of particular significance. However, this finding is only loosely connected to the other findings of Uhrf1 in mESC differentiation and the suggested mechanism of a Uhrf1/Setd1a/COMPASS complex. Further studies of how Uhrf1 affects the binding of the core pluripotent factors in somatic cell reprogramming and mediates active open chromatin conformation (beyond the suggestion that H3K4me3 reduction is associated with Oct4 binding loss in Figure 5F) to promote Oct4 binding would be beneficial.

Overall, while there are certainly interesting and suggestive results presented concerning the role of Uhrf1 in somatic cell reprogramming, the conclusions made, particularly in the first 3 Figures of the paper, require substantial experimental support.

Major Concerns:

1) The majority of Figure 1 is focused on quantitative analysis of ChIP-seq experiments, but it does not appear any controls were used to allow for quantitative comparison between ChIPs. Therefore, it is difficult to distinguish between reductions and nucleosome redistribution. Spiking in an alternative genome (such as from yeast or drosophila), would provide some level of normalization and comparison between data sets. Deeper reads (30-50 million reads for each ChIP-seq rather than the 8-10 million presented for H3K4me3 and H3K27me3) are also needed in order to make quantitative conclusions, while input reads should be equal or double the number of ChIP-seq reads (rather than less as presented). Also, no validation data for these antibodies is presented or described. And where is the related 5mc and 5hmc data from in Figure S1A?

2) The proposed mechanism and many of the experiments linking Uhrf1 and the Setd1a/COMPASS complex hinge upon the MS findings in Figure 3B. However, these findings appear weak at best as the cutoff for unique peptide sequences for each protein was set very low in order to include Setd1a, Ash2l, and Mll2. Uhrf1 itself has only 14 peptides and is not represented in the top 5 hits. This suggests the MS dataset is of poor quality. Plus, Rbbp4 of the PRC2 complex is represented at low but much higher levels than the components of the Setd1a complex. Furthermore, the findings do not appear to agree with other Uhrf1 MS datasets, in which Dnmt1 should be much more strongly represented along with Usp7, PCNA, and other proteins (see Biogrid.com for a comprehensive list of identified interactions from MS experiments). How many times was this replicated? Were the results consistent? What do you make of the other peptides represented at much higher levels than those focused on in Figure 3B?

3) Co-IP binding experiments with Uhrf1 domain deletions (Figure S3C) are not convincing. Protein levels were unequal between the different Uhrf1 mutants, which makes it hard to interpret pulldowns and appears to suggest that the TTD domain (mis-labeled as the TDD domain throughout Figure S3) is required more than the SRA domain for Setd1a binding.

4) It is necessary to show that Setd1a and components of the complex are present/were pulled down with Uhrf1 in the IP if it is being proposed that the Setd1a/COMPASS complex binds Uhrf1 and is responsible for H3K4 methylation. (Figure 3D) Otherwise, what is the histone methyltransferase in Line 273?

5) Figure 3E is a key experiment to support the proposed Uhrf1/Setd1a complex. The data presented is not convincing.

6) Another major premise of this paper is that Uhrf1 is important for specifying the neuroectodermal lineage through regulation of H3K4me3 levels. However, the effect of Uhrf1 on levels of this post-translational modification is only directly shown *in vivo* in MEFs (Figure S3A), which are already fate-restricted towards the mesodermal lineage. The effect of Uhrf1 overexpression and knockdown on H3K4me3 would be more meaningful and relevant towards the main hypothesis if repeated in ESCs. Also, this important western blot requires Uhrf1 immunoblotting and a cleaner H3 blot to discern the subtle change in H3K4me3.

Minor Concerns:

1) Why is there such a strong focus on the ability of Uhrf1 to promote the neuroectodermal lineage through H3K4me3? The data just as equally suggests that instead Uhrf1 represses the mesodermal lineage through regulating H3K27me3 levels. This, along with the missing experiments above, suggest the title does not accurately reflect the data presented in this paper.

2) Many grammatical errors and spelling mistakes make the paper difficult to read. Also, there are inconsistencies in citation format (Line 75) and concerns about the appropriateness of a number of citations used in the introduction. For example, three of the first four citations are in reference to zebrafish. More relevant mouse and human references should be included.

3) In Figure 1B, these are mouse KO ESCs, correct? They are labeled DNMT1 KO in the figure.

4) It is hard to see differences between WT and Uhrf1 KO embryoids in the images presented in Figure 2A.

5) Figure 2B suggests that embryoid "beating" started early in the absence of Uhrf1 and decreased by Week 2. Is a corresponding change in cell-type composition seen in the immunostaining of differentiated embryoids in Figure 2E? (Also, the figure says this immunostaining was at Week 1 but the legend says it was Day 14.)

6) The proteins in Figure 3B are color-coded incorrectly.

7) Figure S4C needs a better blot. Where is Uhrf1 in the first column?

Reviewer #3 (Remarks to the Author):

In their manuscript "Active transcription regulation by Uhrf1 is essential for neuroectodermal specification and somatic cell reprogramming" the authors present a number of interesting findings on additional functions of Uhrf1 besides its well-documented role for DNA methylation maintenance. Overall the experiments are well done and quite encompassing. The authors investigate the function of Uhrf1 for multiple different cell states and conversion processes (pluripotent stem cells, differentiation into the germ layers and reprogramming of somatic cells to pluripotency). Their interaction studies were convincing and their genome-wide analysis (ChIP-Seq for H3K4me3, H3K27me3, Oct4 and ATAC-Seq) showed robust changes in Oct4 binding and the

overall epigenetic profile upon loss of Uhrf1 in mESCs. Studies such as this are necessary to emphasize the often complex roles for epigenetic regulators such as Uhrf1.

On the other hand, there are also major caveats that limit the reviewer's enthusiasm.

1. Most data are from one wild-type and one Uhrf1 KO line. The differences observed in ES cell differentiation and epigenetic states could be due to line-to-line differences rather than the effect of the Uhrf1 KO. The authors should perform a rescue experiment by expressing Uhrf1 transgene in the KO line, or generate a new KO line, and verify all key findings reported in this paper.

2. DNA methylation status must be examined either genome-wide or at least at several loci of critical importance to the paper. Uhrf1 KO cells are expected to have global methylation changes. Differences in DNA methylation could be responsible for some if not all of the differentiation and reprogramming defects observed in Uhrf1 KO cells. The difference in histone modification could also result from a DNA methylation defect.

3. Many figures throughout the manuscript don't show statistic analysis. For instance, a main conclusion of the paper is that Uhrf1 deficiency causes neural differentiation defects, but Figure 2C doesn't show significant difference between WT and Uhrf1 KO in differentiation to any of the germ layers. In a number of cases, it's unclear what are the exact "controls" used. All Western blots need to indicate how many times the experiments were repeated.

Additional Points:

1. A number of key points need to be explained or clarified.

- Overall there is low enrichment of Uhrf1 (Fig 1A) at promoters, yet loss of Uhrf1 has a dramatic effect on promoter methylation marks. Can the authors explain this apparent discrepancy?

- Can the authors explain why the H3K4me3 mark at active promoters is not affected as it is in bivalent promoters after Uhrf1 KO?

- The authors primarily focused on bivalent promoters that lost the H3K4me3 mark (13.95 % of bivalent promoters) but they note that 33.57% of bivalent promoters lost the H3K27me3 mark but retain H3K4me3 (page 5). Are these "activated" bivalent promoters enriched for particular kinds of genes? Do they show increased expression in mESCs (undifferentiated or upon differentiation)? Could they explain the bias of Uhrf1 KO mESCs towards mesoderm and away from ectoderm?

- "Around half of Oct4 binding sites in WT were replaced with Sox2 or Nanog in KO, and the rest were lost in KO (Figure 5C)." The authors should clarify the meaning of "replacement". It is known that most Oct4 bound sites are co-bound by Sox2 and Nanog. Do the authors mean a switch from the binding by all three factors, to the binding by the two or one of the remaining two factors? Or do the authors mean there's a gain of ectopic Sox2 and/or Nanog binding in KO cells at sites bound by Oct4 alone in wt cells?

2. On page 10, "Genes involved in stem cell maintenance were directly targeted by replacement of Sox2 and Nanog (Figure S5A) but fig S5A legend says "Enrichment of genes related to stem cell maintenance in target genes by loss and replacement of Oct4 binding". So are the data on sites with Oct4 replacement only, or both replacement and loss. The bigger question is what is this enrichment compared to. If it's compared to any sites, it is known that Oct4 bound genes are enriched with genes related to stem cell maintenance, so it's not surprising that a subset of Oct4 bound genes are also enriched with such genes.

3. Fig 1C, y axis needs to be properly labeled. There's no Figure S1H. The authors probably meant S1G on page 5.

4. Figure 2. For Fig. 2D the x-axis and y-axis is cut off for some of the graphs. Also it is necessary to perform significance tests. At the moment the message of this figure is hard to understand. Fig 2G need to provide error bars for the wild-type teratoma size. Teratomas are notoriously variable

in size. Having a sample size of $n=2$ is certainly not sufficient to conclude a significant difference between the teratomas.

5. Fig S2B, it's unclear whether the retention rate is calculated by comparing KO to WT at each time point, or comparing KO cells at each time point to undifferentiated KO or WT cells. Fig S2E, It's unclear what exactly the graph is quantifying (how many sections, how many teratomas for example). Teratoma samples are highly variable, and tissue organization within a teratoma is also heterogeneous. The lineage bias observed here may not be reproducible.

6. Figure 3. The majority of experiments presented in Fig 3 and the corresponding Fig S3 were performed using protein complexes purified from human 293T cells, rather than mouse ES Cells. These experiments should be repeated using protein complexes purified from mouse ES cells. There are also many issues with the data shown in Fig 3.

Fig. 3B the color codes in the legend are switched for polycomb complex and TRX/MLL complex. For Fig 3C is there a negative control (western blot for a protein that does not interact with Uhrf1). The IgG control alone is not sufficient. Fig 3E is not convincing at all. The shRNA vector control also didn't show significant H3K4me3. Also the data in Fig 3D, 3E and 3F are not consistent. Fig 3D showed a 2,000 fold fold change, whereas the 3E and 3F showed only 2-5 fold changes with Uhrf1 transfection. Also Setd1a in the top labels (for the columns) is misspelled. Fig 3F is also problematic. Although SRA deletion does appear less efficient in inducing H3K4me3, the corresponding Uhrf1 protein level is also greatly reduced as shown in Fig S3C.

7. Fig S3A. From the image, H3K27me27 and H3K9me3 also have a positive correlation with the expression of Uhrf1. However, there are a few issues. First of all, the expression level of Uhrf1 should be examined by Western. Secondly, how many times was the experiment performed, and quantification of the Western bands should be provided for statistic analysis.

8. Figure 4. For Fig 4C. Please provide flow data for surface markers such as SSEA-1. Immunofluorescence images should be provided for pluripotency transcription factors (Oct4, Nanog, Sox2). For people to understand Fig 4D, it is necessary to explain the coat color and the background of the MEFs used for making the iPS cells.

9. Fig S4C is not convincing. The Uhrf1 protein expression is barely detectable. What are the exact controls used in Fig 4E and Fig S4E?

10. Fig 5. The reduction in Oct4 binding could be due to a reduction in Oct4 protein expression. Oct4 protein expression in WT vs KO should be evaluated by Western. Fig 5D, it's understandable that regions that lost Oct4 binding also showed reduced chromatin accessibility, but why would regions that had replacement of Oct4 binding with Sox2 and/or Nanog binding show increased chromatin accessibility?

Response to Reviewers' comments:

We truly appreciate the valuable comments from reviewers. Below, please see below the response to comments in blue, and manuscript with changes in blue.

Reviewer #1 (Remarks to the Author):

Uhrf1 plays critical roles in maintaining DNA methylation and depositing histone H3K9me3 by recruiting Dnmt1 and KAP1/Setdb1, respectively. In this study, Kim et al. showed that Uhrf1 interacts with the Setd1/COMPASS complex and is essential for neuroectodermal specification and somatic cell reprogramming. These findings are potentially significant. However, the proposed mechanisms are not entirely supported by convincing experimental data.

Major issues:

1. It has been shown that Uhrf1 binding to hemimethylated DNA can inhibit Setdb1-mediated H3K9me3 deposition at some loci (Sharif et al., Cell Stem Cell 2016). Given that H3K4me3 and H3K9me3 are often antagonistic, it is possible that the decrease in H3K4me3 in Uhrf1-deficient cells is, at least in part, a secondary effect of H3K9me3 increases. The authors ought to conduct experiments to exclude that possibility or determine the relative significance of the Setd1/COMPASS and KAP1/Setdb1 complexes.

We appreciate reviewer's comment. In order to evaluate whether the decrease of H3K4me3 in Uhrf1 KO is responsible for the increase of H3K9me3 marks, we compared H3K4me3 marks between WT and Uhrf1 KO on differential H3K9me3 regions (below Figure). We found that the loci of H3K9me3 gain in Uhrf1 KO (384, Green in below Figure) did not show difference in H3K4me3. Likewise loci of H3K9me3 loss in Uhrf1 KO (4,524, Red in below Figure) also do not show a difference in H3K4me3. These results indicate that a decrease in H3K4me3 in Uhrf1 KO cells was not due to the loss of antagonistic effect of H3K9me3. Please see the below figure.

2. Some results are inconsistent. For example, the effect of FLAG-Uhrf1 is ~2000 fold over vector control in Fig. 3D, whereas it is only ~2.5 fold in Fig. 3E (sc-con vs. Vector con). Is it because of FLAG-Uhrf1 expression levels, cell type difference (HEK293 vs ES), or experimental variations? The blot and the bar graph in Fig. 3E are also inconsistent. Specifically, both the Vector control and sc-con samples show a faint signal in the H3K4me3 blot and do not seem to be different.

We appreciate reviewer's comment. Figure 3D presented the absolute read in HMT assay after subtracting the background, while other Figures (Figure 3E and 3F) present the data normalized against vector control. Thus, Figure 3D seemed to show a lot more change in HMT activity. When normalized against vector control, Figure 3D shows a similar fold change like shown below. In revising the manuscript, we put new HMT results that were performed with mESCs (Figure 3D).

For the Figure 3E, we repeated the experiments, and present the data for H3K4me3 showing clear difference (new Figure 3E).

3. The authors claimed that the SRA domain mediates Uhrf1 interaction with Setd1a. Several issues need to be clarified: a) Is the interaction direct? b) Biochemical and structural data indicate that SRA domain binds DNA. Is Uhrf1-Setd1a interaction dependent on the presence of DNA? c) From Fig. S3C, deletion of UBL or TTD also seems to affect (albeit not abolish) the interaction. d) The methylation data in Fig. 3F are inconsistent with the mapping data in Fig. S3C.

We appreciate reviewer's insightful comments.

In addressing the comment a) and b), whether Uhrf1 and Setd1a directly interact each other, and/or whether the interaction is DNA dependent, we perform the *in vitro* binding assay with recombinant proteins purified from E.coil. The large size of Setd1a (1716 AA, over 250 kD) made it challenging to purify full length Setd1a. So, we first identified the domain of Setd1a that mediates the interaction with Uhrf1. Using a series of deletion mutants of Setd1a and Uhrf1, we found that SET domain of Setd1a and SRA domain of Uhrf1 are essential for their interaction. This data is presented as Figure 3H. After identification of the interacting domains, we purified full-length recombinant Uhrf1 and obtained the recombinant partial SETD1A, a human homologue of Setd1a (amino acid 1418 – end, Creative BioMart, SETD1A-198H) that has catalytic N-SET and SET domain. Using the recombinant proteins, we found that purified full length Uhrf1 and partial SETD1A directly interact in *in vitro* condition, without DNA or other previous known Setd1a/COMPASS complex proteins. This data is presented as Figure S4F. In order to further confirm the DNA-independency for interaction, we performed the Co-IP experiments with DNase I treatment. The DNase I treatment did not abolish the Uhrf1-Setd1a interaction, supporting that Setd1a and Uhrf1 interact each other with DNA. This data is presented as Figure 3F.

In addressing comment c), whether PHD and TTD domains are also important for interaction with Setd1a, we repeated the Co-IP experiments and HMT assay multiple times. We confirmed that PHD and TTD has minimal role in mediating interaction with Setd1a. New data for interaction is presented as Figure 3F.

In addressing comment d), whether PHD or TTD domains are also important for H3K4me3 activity in addition to SRA domain, we distinguished the HMT activity toward H3 or chromatin as substrates. When we performed HMT assay using H3 as substrate, we found that Uhrf1 deleted of PHD or TTD domains did not show the decrease of activity (Figure 3G). However, when chromatin was used as substrate, Uhrf1 mutants deleted of PHD, TTD or SRA showed the diminished HMT activity (Figure S4E). We think that the difference in activity toward H3 or chromatin substrate is due to the fact that PHD or TTD domains of Uhrf1 are important in mediating methyltransferase activity toward DNA or histones other than H3. Although we have not further pursued what are the substrates whose methylation is affected by PHD or

TTD deletion, our data strongly support that SRA domain of Uhrf1 is necessary and sufficient in binding with Setd1a and mediating H3K4me3.

4. The authors showed that Uhrf1^{-/-} MEFs failed to be reprogrammed by Yamanaka factors (Fig. 4E). Do these cells show defects in survival and proliferation compared to WT MEFs?

We measured the proliferation of Uhrf1^{-/-} MEF and found that there is no difference of proliferation compared with wild type MEF. The data is added as Figure 4E.

Minor issues:

1. A Uhrf1 blot is needed to show overexpression and the knockdown efficiency in Fig. S3A.

We performed Western blot for Uhrf1 to show the overexpression and knockdown efficiency against Uhrf1 in MEFs as well as mESCs (now new Figure S4A).

2. Page 5, lines 148-151. The authors described that more than half of repressive marks and bivalent marks in WT ESCs were lost in Uhrf1^{-/-} ESCs. Yet, their total fractions showed no obvious changes (Fig. 1D). Does that mean Uhrf1^{-/-} ESCs also gain repressive marks and bivalent marks in other genomic regions? Please clarify.

We really appreciate reviewers' comments. We carefully re-analyzed the data.

We observed that 1,124 out of 1,693 repressive promoters were switched into other marks by Uhrf1 KO. However, 1,194 repressive promoters were also gained from the other marks (Figure 1D). Therefore, total number of single repressive marks did not show much difference between WT and KO cells (Figure S1E, Blue). In contrast, loss of bivalent marks were detected in 1,956 promoters, whereas the gain in 552 promoters, showing a decrease of total number of bivalent promoters in Uhrf1 KO mESCs. We described details in the main text, and we replaced the previous Figure 1D with Figure S1E.

3. Fig. 2F. It would be helpful to define B-A, B-R, R-N in the figure legend.

We described the definition of them in the figure legend.

4. Fig. 5G. It would be helpful to add scale bars (in kb) to indicate the relative locations of

binding peaks and genes.

We added scale bars in Fig 5G.

5. TTD (tandem Tudor domain) was misspelled as TDD throughout the manuscript.

We apologize the misspelling. We now have replaced TDD with TTD in Figures, legends, and text.

6. Fig. 3F. shSetd1a was misspelled as shSeta1a. In Fig. S3F, shSet1A should be changed to shSetd1a to be consistent.

We now have corrected Setd1a throughout Figures and legends.

7. Fig. S2B. The x-axis should be indicated (weeks).

We added "weeks" at x-axis.

8. Figure S1H cited on Page 5, line 167 was not found in Figure S1.

We mistook Figure number. This should be Figure S1G. We corrected the figure number.

9. Fig. S2C, lower panel. The labeling (H3K4me3, etc.) and the data were not aligned properly.

We correctly aligned the data label.

Reviewer #2 (Remarks to the Author):

Uhrf1 plays an important role in the inheritance of DNA methylation patterns during DNA replication. Despite the close relationship between Uhrf1 and accurate maintenance of these patterns, little is known about how Uhrf1 may influence pluripotency-specific epigenetic states. In this manuscript, Kim et al., suggest Uhrf1 helps maintain pluripotency and stabilize the mouse ESC epigenome through modulating active histone marks at bivalent loci. They further suggest this is through an interaction with the Setd1a/COMPASS complex to promote H3K4me3 and the neuroectodermal lineage such that in the absence of Uhrf1, neuroectodermal lineage fate-restriction is limited. Additionally, Kim et al., show Uhrf1 is necessary for somatic cell reprogramming, particularly through recruitment of Oct4 to binding sites.

The main conclusion of this manuscript, that Uhrf1 regulates the neuroectodermal lineage through an interaction with the Setd1a complex to catalyze H3K4me3, is not convincingly supported. Key experiments supporting this model are hard to interpret (detailed below). Furthermore, despite the focus on the neuroectodermal lineage, Figures 1-3 show conversely (and perhaps more strongly) that Uhrf1 is important for repressing the mesodermal lineage and promoting H3K27me3. (Figure 1C middle panel, Figure S1A Plot 1 vs Plot 2, Figure S1B, Figure 2B-E, etc.), suggesting that promoting neuroectodermal fate-restriction is not the key mode of action of Uhrf1 in determining mESC pluripotency.

Figures 4 and 5 present highly interesting and suggestive data concerning Uhrf1's role in somatic cell reprogramming; this data could be very important for our understanding of molecular mechanisms regulating iPSC reprogramming. The evidence showing Uhrf1 is almost as efficient at promoting MEF reprogramming as c-Myc is of particular significance. However, this finding is only loosely connected to the other findings of Uhrf1 in mESC differentiation and the suggested mechanism of a Uhrf1/Setd1a/COMPASS complex. Further studies of how Uhrf1 affects the binding of the core pluripotent factors in somatic cell reprogramming and mediates active open chromatin conformation (beyond the suggestion that H3K4me3 reduction is associated with Oct4 binding loss in Figure 5F) to promote Oct4 binding would be beneficial.

Overall, while there are certainly interesting and suggestive results presented concerning the role of Uhrf1 in somatic cell reprogramming, the conclusions made, particularly in the first 3 Figures of the paper, require substantial experimental support.

Major Concerns:

- 1) The majority of Figure 1 is focused on quantitative analysis of ChIP-seq experiments, but it does not appear any controls were used to allow for quantitative comparison between ChIPs. Therefore, it is difficult to distinguish between reductions and nucleosome redistribution. Spiking in an alternative genome (such as from yeast or drosophila), would

provide some level of normalization and comparison between data sets. Deeper reads (30-50 million reads for each ChIP-seq rather than the 8-10 million presented for H3K4me3 and H3K27me3) are also needed in order to make quantitative conclusions, while input reads should be equal or double the number of ChIP-seq reads (rather than less as presented). Also, no validation data for these antibodies is presented or described. And where is the related 5mC and 5hmC data from in Figure S1A?

We appreciate reviewer's suggestions. We carefully have considered the possibility of spike-in and the effect of read depth. The utilizing of constant amount of spike-in epigenome may be helpful to remove biases among multiple ChIP samples. However, spiking approach in ChIP-seq has not been extensively evaluated yet. There are several issues including balancing the amount of spike-in (Meyer et al, Nature Review Genetics, 2014) and an unexpected decrease in the read density in the background (Nakato et al., Briefings in Bioinformatics, 2017), which should be solved for the accurate quantification.

Higher read depth may be one of factors to determine the reliability of sequencing data. To investigate the effect of read depth on our quantification analysis, we compared our H3K4me3 ChIP-seq data with that from ENCODE project, which used the same E14 cell line and same antibody (04-745, Millipore) with higher amount of reads and double number of input. Despite the difference of read count, we found that H3K4me3 level ($\log_2(\text{ChIP}/\text{input})$) around TSS was significantly correlated between our and ENCODE data (See left figure). In addition, more than 90% of active promoters were overlapped between our and ENCODE data (See right figure), indicating that the read count in our datasets does not show significant effect on our quantification.

We also note that antibodies used for ChIP-seq has widely used and validated by many groups (e.g. Yang et al, 2017, Illingworth et al., 2016, Kidder et al., 2017, King et al., 2017). For 5hmC and 5mC, we used public (h)MeDIP-seq data. We described the information of the source and data processing of (h)MeDIP-seq in Method section.

2) The proposed mechanism and many of the experiments linking Uhrf1 and the Setd1a/COMPASS complex hinge upon the MS findings in Figure 3B. However, these findings appear weak at best as the cutoff for unique peptide sequences for each protein was set very low in order to include Setd1a, Ash2l, and Mll2. Uhrf1 itself has only 14 peptides and is not represented in the top 5 hits. This suggests the MS dataset is of poor

quality. Plus, Rbbp4 of the PRC2 complex is represented at low but much higher levels than the components of the Setd1a complex. Furthermore, the findings do not appear to agree with other Uhrf1 MS datasets, in which Dnmt1 should be much more strongly represented along with Usp7, PCNA, and other proteins (see Biogrid.com for a comprehensive list of identified interactions from MS experiments). How many times was this replicated? Were the results consistent? What do you make of the other peptides represented at much higher levels than those focused on in Figure 3B?

We appreciate valuable reviewer's comments. We have considered the comment very carefully, re-analyzed our MS datasets and perform additional Co-IP validation experiment to confirm that our MS data is of reliable quality. MS experiments with SA or FLAG from BirA-Uhrf1 includes several background noises, which may be derived from high protein expression or affinity with SA or FLAG antibody. Therefore, the number of unique peptide does not directly represent the strength of interactions, and must be normalized by control experiment. We calculated the relative strength of interaction with dividing the number of peptide from BirA-Uhrf1 by that from BirA control. After the normalization, we found that Uhrf1 is ranked as top. We added log₂(ratio) column in Table S1.

We performed MS experiment with five independent experiments in SA and FLAG pulldown. Our MS data identified total 1,264 proteins. We preprocessed our MS data by Wang et al., 2006 criteria with a small modification to remove background proteins. Finally, we narrowed down 232 candidate interaction partners (Table S1). To validate our criteria of MS data, we tested protein-protein interaction with Co-IP experiment. Although several proteins were already tested (Figure 3C), we performed additional Co-IP experiment including proteins (ZNF451 CCAR1, DNAJC2), whose interaction strength is similar with Setd1a. All of them significantly interact with Uhrf1, supporting that the reliability of our MS datasets. Please see the Co-IP data below.

BioGrid database deposits protein-protein interaction from multiple experimental approaches and ranks the interactions by the number of evidences, which represents their reliability but not the strength of interactions. In addition, most of mouse Uhrf1 interaction partners in BioGrid were obtained from non-MS low-throughput approach (co-localization, Affinity Capture-Western). Therefore, it is difficult to compare the peptide count in our high-

throughput MS experiment with BioGrid comprehensive list.

3) Co-IP binding experiments with Uhrf1 domain deletions (Figure S3C) are not convincing. Protein levels were unequal between the different Uhrf1 mutants, which makes it hard to interpret pulldowns and appears to suggest that the TTD domain (mis-labeled as the TDD domain throughout Figure S3) is required more than the SRA domain for Setd1a binding.

We appreciate the valuable comments. In order to identify the Uhrf1 domain that interacts with Setd1a, we repeated the Co-IP experiments more than four times, and confirmed that only SRA domain is essential for interacting with Setd1a. We measured the protein concentration and carefully loaded the equal amount of Co-IP samples for Western blot analysis. We present the new Figure 3F. Furthermore, we used recombinant Uhrf1 and SETD1A to demonstrate that they directly interact (new Figure S4F). Additionally, we used a series of mutants of Setd1a deleted of different domain and found that SET domain of Setd1a is critical for interacting with Uhrf1 (new Figure 3H).

Throughout the manuscript, we corrected typos to replace TDD with TTD.

4) It is necessary to show that Setd1a and components of the complex are present/were pulled down with Uhrf1 in the IP if it is being proposed that the Setd1a/COMPASS complex binds Uhrf1 and is responsible for H3K4 methylation. (Figure 3D) Otherwise, what is the histone methyltransferase in Line 273?

We appreciate the comments. Co-IP experiments of Uhrf1 in Figure 3C demonstrate that the Setd1a/COMPASS complex proteins are present in Uhrf1 complex. Dpy30, Wdr5, Rbbp5, Hcfc1 and Cfp1 that are shown in Uhrf1 Co-IP (Figure 3C) are the main components of Setd1a/COMPASS complex. Proteomic data (Figure 3B) also support the interaction of Uhrf1 with Setd1a/COMPASS complex.

5) Figure 3E is a key experiment to support the proposed Uhrf1/Setd1a complex. The data presented is not convincing.

We appreciate the comments. We agree the importance of Figure 3E, and performed multiple HMT experiments using H3 or chromatin as substrates. Please refer to the answer #3 above of reviewer #1's comments. We present new assay data for H3K4me3 as Figure 3E.

6) Another major premise of this paper is that Uhrf1 is important for specifying the neuroectodermal lineage through regulation of H3K4me3 levels. However, the effect of Uhrf1 on levels of this post-translational modification is only directly shown in vivo in MEFs (Figure S3A), which are already fate-restricted towards the mesodermal lineage. The effect of Uhrf1 overexpression and knockdown on H3K4me3 would be more meaningful and relevant towards the main hypothesis if repeated in ESCs. Also, this important western blot requires Uhrf1 immunoblotting and a cleaner H3 blot to discern the subtle change in H3K4me3.

We appreciate valuable comments. We performed western blot for various histone marks in mESCs. Like in MEFs, Uhrf1 expression level affects the H3K4me3 modification level in mESCs. We added the results together with quantification and statistical analysis in Figure S4A.

Minor Concerns:

1) Why is there such a strong focus on the ability of Uhrf1 to promote the neuroectodermal lineage through H3K4me3? The data just as equally suggests that instead Uhrf1 represses the mesodermal lineage through regulating H3K27me3 levels. This, along with the missing experiments above, suggest the title does not accurately reflect the data presented in this paper.

We agree with reviewer. Our data about the repressive role of Uhrf1 on mesoderm lineage through H3K27me3 is very interesting and equally important as much as a role of Uhrf1 in H3K4me3 mediated neuroectodermal lineage specification. We made this importance in detail in main text (lines 220 – 241, 372 - 384). In line with this, we made a change in title of the study “Active transcription regulation by Uhrf1 is essential for bivalent modification in pluripotency.

2) Many grammatical errors and spelling mistakes make the paper difficult to read. Also, there are inconsistencies in citation format (Line 75) and concerns about the appropriateness of a number of citations used in the introduction. For example, three of the first four citations are in reference to zebrafish. More relevant mouse and human references should be included.

We appreciate the comments. We corrected the citation format and replaced references with those for describing the function of Uhrf1 in mammalian system, including mESC, NSCs, and human cell lines (Reference 5, 6, 7,15).

3) In Figure 1B, these are mouse KO ESCs, correct? They are labeled DNMT1 KO in the figure.

We appreciate the comments. They are data from KO mESCs. We added KO mESCs in the label for clarification.

4) It is hard to see differences between WT and Uhrf1 KO embryoids in the images presented in Figure 2A.

We appreciate the comments. We replaced the old figures with those having wider view. We also added arrow to indicate the cystic hollow in Uhrf1 KO EBs (Figure 2A).

5) Figure 2B suggests that embryoid “beating” started early in the absence of Uhrf1 and decreased by Week 2. Is a corresponding change in cell-type composition seen in the immunostaining of differentiated embryoids in Figure 2E? (Also, the figure says this immunostaining was at Week 1 but the legend says it was Day 14.)

Yes. The immunostaining data in Figure 2E are from EBs from those shown “beating” phenotypes in Figure 2B. When performing the differentiation experiments, some EBs were taken from each week up to week 3 for immunostaining and qPCR. We apologize for the discrepancy in describing the Figure 2E in Text and Figure legend. All the Immunostaining data are from 1 week EBs. We corrected Figure legend accordingly.

6) The proteins in Figure 3B are color-coded incorrectly.

We corrected color code of Figure 3B.

7) Figure S4C needs a better blot. Where is Uhrf1 in the first column?

We performed experiments again and added new one having a clear Uhrf1 band (Figure S5C).

Reviewer #3 (Remarks to the Author):

In their manuscript “Active transcription regulation by Uhrf1 is essential for neuroectodermal specification and somatic cell reprogramming” the authors present a number of interesting findings on additional functions of Uhrf1 besides its well-documented role for DNA methylation maintenance. Overall the experiments are well done and quite encompassing. The authors investigate the function of Uhrf1 for multiple different cell states and conversion processes (pluripotent stem cells, differentiation into the germ layers and reprogramming of somatic cells to pluripotency). Their interaction studies were convincing and their genome-wide analysis (ChIP-Seq for H3K4me3, H3K27me3, Oct4 and ATAC-Seq) showed robust changes in Oct4 binding and the overall epigenetic profile upon loss of Uhrf1 in mESCs. Studies such as this are necessary to emphasize the often complex roles for epigenetic regulators such as Uhrf1.

On the other hand, there are also major caveats that limit the reviewer’s enthusiasm.

1. Most data are from one wild-type and one Uhrf1 KO line. The differences observed in ES cell differentiation and epigenetic states could be due to line-to-line differences rather than the effect of the Uhrf1 KO. The authors should perform a rescue experiment by expressing Uhrf1 transgene in the KO line, or generate a new KO line, and verify all key findings reported in this paper.

We appreciate reviewer’s comments. In order to exclude the possibility that what we observed was due to the line-to-line variation, we generated another Uhrf1 K/O ESC from Uhrf1 fl/fl mouse (B6Dnk;B6N-Uhrf1^{tm1a(EUCOMM)Wtsj}/leg). We described the derivation in materials and methods section. From embryos of Uhrf1 fl/fl mouse, Uhrf1 fl/fl mESC lines were generated, one of which is JES6. After expressing Cre recombinase, we deleted Uhrf1 (JESC6-Cre), and used for ES differentiation experiments. JES6-Cre Uhrf1 KO ESC showed abnormal EB beating and defect in neuro-ectodermal lineage differentiation like the original Uhrf1 KO ESC. New data is presented as Figure S3A-E.

Furthermore, we performed the rescue experiment. We stably introduced the Uhrf1 into Uhrf1 KO ESC and induced EB differentiation. Abnormal phenotypes of observed in Uhrf1 KO line, including EB beating and defect of neural differentiation, were rescued with Uhrf1 expression, further confirming that these phenotypes are due to the loss of function of Uhrf1 but not the line-to-line variation. We added the result in Figure 2A, 2B, and 2C.

2. DNA methylation status must be examined either genome-wide or at least at several loci of critical importance to the paper. Uhrf1 KO cells are expected to have global methylation changes. Differences in DNA methylation could be responsible for some if not all of the differentiation and reprogramming defects observed in Uhrf1 KO cells. The difference in histone modification could also result from a DNA methylation defect.

The crosstalk of DNA methylation and histone modification has been reported by many physical interactions of histone modification enzymes and DNA methyltransferases (Cedar et

al., 2009). Since Uhrf1 also acts as a link between H3K9me3 and maintenance of DNA methylation via interaction with Dnmt1 (Cheng et al., 2013, Hashimoto et al., 2010 and Rajakumara et al., 2011), it is very difficult to exclude the effect of DNA methylation on differentiation and reprogramming completely. However, others and we have shown many evidences that Uhrf1 KO or overexpression display distinct phenotypes in differentiation and reprogramming from Dnmt1 KO or overexpression. For example, ectodermal genes were downregulated in Uhrf1^{-/-} EBs (Fig. 2), but upregulated in Dnmt1^{-/-} EBs (Schmidt et al., 2012). Although we demonstrated the enhancement of iPSC generation by Uhrf1 overexpression, inhibition of Dnmt1 accelerates the gain of pluripotency (Mikkelsen et al., 2008). We also found that Dnmt1 OE has little effect on iPSC reprogramming. Please see the below AP staining images. In addition, genome-wide Uhrf1 distribution was significantly correlated with histone modifications than DNA methylation (Fig. S1A). Overall, our results indicate a limited effect of DNA methylation on Uhrf1-mediated differentiation and reprogramming.

We also analyzed the relationship of loss of DNA methylation by Uhrf1 KO with histone modification changes. As reported previously (Cheng et al., 2013, Hashimoto et al., 2010 and Rajakumara et al., 2011), we observed significant correlation between DNA methylation and H3K9me3 change (Cor=0.119 & p=3.11e-15). H3K27me3 is also significantly correlated with DNA methylation change (Cor=0.104 & 3.12e-15). However, DNA methylation change is not associated with H3K4me3 change (Cor=0.0058, p=0.575). At least, H3K4me3 change by Uhrf1 is DNA methylation-independent mechanism.

3. Many figures throughout the manuscript don't show statistic analysis. For instance, a main conclusion of the paper is that Uhrf1 deficiency causes neural differentiation defects, but Figure 2C doesn't show significant difference between WT and Uhrf1 KO in differentiation to any of the germ layers. In a number of cases, it's unclear what are the exact "controls" used. All Western blots need to indicate how many times the experiments were repeated.

We appreciate the comments. As for Figure 2C, we added the information to the statistical analysis. Additionally, we derived and used new ESC line for Uhrf1 KO to repeat the EB differentiation experiments. Furthermore, we performed rescue experiments by introducing Uhrf1 into Uhrf1 KO ESC and performing EB differentiation experiments. The data is presented in Figure 2A-C. As for the description of the data, we compared the expression level of genes representing the three germ layers between wild type and Uhrf1 KO EBs using qPCR and normalized the value with wild type control mESCs at day 0 before differentiation. Compared with wild type EBs, Uhrf1 KO EBs showed statistically significant increase of mesodermal markers (Nkx2.5 and Bmp5) at earlier stage of differentiation and showed decrease in expression of neuroectodermal markers (Nestin and TuJ).

According to the comments, throughout the manuscript we indicated how many experiments were performed and added statistical analysis.

Additional Points:

1. A number of key points need to be explained or clarified.
 - Overall there is low enrichment of Uhrf1 (Fig 1A) at promoters, yet loss of Uhrf1 has a dramatic effect on promoter methylation marks. Can the authors explain this apparent discrepancy?

Our ChIP-seq for Uhrf1 indicates that Uhrf1 was globally distributed in both intra- and intergenic regions (about 5% of genome is protein-coding regions in mouse). Our results demonstrate that loss of Uhrf1 affects not only promoter marks (H3K4me3 and H3K27me3), but also intergenic marks (H3K9me3) (Fig. S1C). Therefore, Uhrf1 regulates different histone modifications of different genomic locations.

- Can the authors explain why the H3K4me3 mark at active promoters is not affected as it is in bivalent promoters after Uhrf1 KO?

When we compared Uhrf1 binding in each promoter type, Uhrf1 enrichment in active promoters was significantly lower than that in the other promoter types (Fig. S1F). Therefore, Uhrf1 KO displays little effect on H3K4me3 mark at active promoters. We added statistical significance of low level of Uhrf1 in active promoters in Fig. S1F.

- The authors primarily focused on bivalent promoters that lost the H3K4me3 mark (13.95 % of bivalent promoters) but they note that 33.57% of bivalent promoters lost the H3K27me3 mark but retain H3K4me3 (page 5). Are these “activated” bivalent promoters enriched for particular kinds of genes? Do they show increased expression in mESCs (undifferentiated or upon differentiation)? Could they explain the bias of Uhrf1 KO mESCs towards mesoderm and away from ectoderm?

We appreciate valuable comments from reviewer. We found that “activated” bivalent promoter significantly occurs in heart developmental genes (mesoderm lineage) (FDR=2.81e-3), whereas “suppressed” bivalent promoter occurs in neuro developmental genes in ESCs (FDR=1.45e-5) (Fig. 2F). However, we did not see significant difference of mesodermal and ectodermal gene expression in WT and Uhrf1 KO mESCs (Figure 1F), because Uhrf1 KO mESCs still maintain stem cell identity with comparable expression of many pluripotent genes with WT. However when we induce the differentiation into 3 germ layers, the heart developmental genes have “activated” bivalent promoter showed dramatic increase and the neuro-developmental genes have “suppressed” bivalent promoter showed defect in gene expression. Therefore, Uhrf1 KO mESCs display the bias of histone

modification pattern toward mesoderm and away from ectoderm and lead into skewed mesodermal lineage differentiation upon EBs differentiation. We described this in main text.

- "Around half of Oct4 binding sites in WT were replaced with Sox2 or Nanog in KO, and the rest were lost in KO (Figure 5C)." The authors should clarify the meaning of "replacement". It is known that most Oct4 bound sites are co-bound by Sox2 and Nanog. Do the authors mean a switch from the binding by all three factors, to the binding by the two or one of the remaining two factors? Or do the authors mean there's a gain of ectopic Sox2 and/or Nanog binding in KO cells at sites bound by Oct4 alone in wt cells?

Reviewer's comment was very helpful to improve our manuscript. In Figure 5C, we compared WT-specific Oct4 binding sites (11,620 sites in Fig 5B) with KO-specific Sox2 (15,203 sites) and Nanog binding sites (7,858 sites). We defined WT-specific Oct4 peaks overlapped with and without KO-specific Sox2 or Nanog peaks as "replacement" and "loss", respectively. Thus, for example, "Sox2+Nanog replacement" means that Oct4 alone in WT cells were changed into Sox2 and Nanog binding in KO cells. "Sox2 replacement" means that Oct4 alone or Oct4+Nanog in WT were changed into Sox2 or Sox2+Nanog binding in KO cells. We have described more details of this analysis in the main text and method section.

2. On page 10, "Genes involved in stem cell maintenance were directly targeted by replacement of Sox2 and Nanog (Figure S5A) but fig S5A legend says "Enrichment of genes related to stem cell maintenance in target genes by loss and replacement of Oct4 binding". So are the data on sites with Oct4 replacement only, or both replacement and loss. The bigger question is what is this enrichment compared to. If it's compared to any sites, it is known that Oct4 bound genes are enriched with genes related to stem cell maintenance, so it's not surprising that a subset of Oct4 bound genes are also enriched with such genes.

We apologized that our description made reviewer confused. Here, we performed GO analysis to target genes of loss or replacement of Oct4 binding and clarified difference of characters between loss and replacement. We found that GO terms related with stem cell maintenance were significantly enriched in replacement of Oct4 ($FDR < 4.96e-4$), but not loss of Oct4 ($FDR = 0.879$). Despite the loss of Oct4 by Uhrf1 KO, Uhrf1KO ESCs still maintain pluripotency and expression level of most of pluripotent genes except for 2C genes were not different between WT and KO. Perhaps Sox2 and Nanog compensate a part of dysregulation of Oct4 to maintain ESC identity. We revised figure legend to make it clear.

3. Fig 1C, y axis needs to be properly labeled. There's no Figure S1H. The authors probably meant S1G on page 5.

We apologize for the missing Y-axis labeling for Figure 1C. We corrected the label. This is now presented as Figure S1G.

4. Figure 2. For Fig. 2D the x-axis and y-axis is cut off for some of the graphs. Also it is necessary to perform significance tests. At the moment the message of this figure is hard to understand. Fig 2G need to provide error bars for the wild-type teratoma size. Teratomas are notoriously variable in size. Having a sample size of n=2 is certainly not sufficient to conclude a significant difference between the teratomas.

We performed statistical tests on Figure 2D and add “*” as significant difference. For the teratoma assay, we increased the number of biological replicates (total n=6, with additional teratoma sets n=4) (Figure 2H). We added error bar with statistical analysis (Figure S2E).

5. Fig S2B, it's unclear whether the retention rate is calculated by comparing KO to WT at each time point, or comparing KO cells at each time point to undifferentiated KO or WT cells. Fig S2E, It's unclear what exactly the graph is quantifying (how many sections, how many teratomas for example). Teratoma samples are highly variable, and tissue organization within a teratoma is also heterogeneous. The lineage bias observed here may not be reproducible.

In Fig S2B, the retention rate was calculated by comparing KO to WT at each time point. We revised our manuscript to describe in more detail (Methods section, Data processing and analysis of RNA-seq, ChIP-seq and ATAC-seq). Original Figure S2E represented the area in teratoma positive for DAB staining for TuJ. Although we observe the overall decrease of TuJ positive region in Uhrf1 KO teratoma, due to the difficulty in quantification, we removed the data.

6. Figure 3. The majority of experiments presented in Fig 3 and the corresponding Fig S3 were performed using protein complexes purified from human 293T cells, rather than mouse ES Cells. These experiments should be repeated using protein complexes purified from mouse ES cells. There are also many issues with the data shown in Fig 3.

Fig. 3B the color codes in the legend are switched for polycomb complex and TRX/MLL complex. For Fig 3C is there a negative control (western blot for a protein that does not interact with Uhrf1). The IgG control alone is not sufficient. Fig 3E is not convincing at all. The shRNA vector control also didn't show significant H3K4me3. Also the data in Fig 3D, 3E and 3F are not consistent. Fig 3D showed a 2,000 fold fold change, whereas the 3E and 3F showed only 2-5 fold changes with Uhrf1 transfection. Also Setd1a in the top labels (for the columns) is misspelled. Fig 3F is also problematic. Although SRA deletion does appear less efficient in inducing H3K4me3, the corresponding Uhrf1 protein level is also greatly reduced as shown in Fig S3C.

We appreciate the critical comment on Figure 3 and S3. We agree with the comments. In order to support the conclusions, we performed the several experiments.

1) We performed HMT data multiples in mESCs as well as in 293T cells. We present the data with mESCs in Figure 3D (n=4).

2) In Figure 3B, we corrected color codes in Figure legend.

3) Figure 3C, when we performed the IP with Uhrf1, ARS2, RBAP46, Menin, and WDR82 did not show an interaction with Uhrf1. Please see the attached results in below.

4) The original Figure 3D presented the absolute read counts in HMT assay after subtracting the background, while other Figures (Figure 3E and 3F) presented data normalized against vector control. Thus, Figure 3D seemed to show a lot more change in HMT activity. When normalized against vector control Figure 3D shows a similar fold change like shown below. In revising the manuscript, we put new HMT results that were performed with mESCs (Figure 3D).

5) For the Figure 3E, we repeated the experiments, and presents with new H3K4me3 western blot showing clear difference (new Figure 3E).

6) In identifying the domains of Uhrf1 important for interacting with Setd1a and HeK4me3 modification, we performed the additional experiments. We performed multiple Co-IP experiments, measured the cell lysates for loading samples for Western blot and confirmed that SRA domain alone is critical interacting with Setd1a (Figure 3F). We also found that SRA domain alone is sufficient to interact with Setd1a (Figure S4C). Furthermore, we found that SET domain of Setd1a is critical for interacting with Uhrf1 (Figure 3H). As for the HMT activity, we performed HMT assay using either H3 or chromatin as substrates. We found that only SRA domain deletion of Uhrf1 decreased the HMT activity toward H3 (Figure 3G). Interestingly, we found that deletion of TTD or PHD domains affected the HMT activity toward chromatin substrate (Figure S4E). Chromatin substrates contain all four core histones (H2A, H2B, H3 and H4) in addition to DNA. Thus, it seems likely that TTD and PHD domains of Uhrf1 are important for modifying the other histones than H3, which is beyond the scope of current manuscript. This data further strengthen our finding that interaction of Setd1a with Uhrf1 mediate the methylation of H3K4. We also performed the in vitro binding assay of Uhrf1 and SETD1A using purified recombinant proteins to show the direct interaction of Uhrf1 with Setd1a (Figure S4F). We revised the text according to the results.

7. Fig S3A. From the image, H3K27me27 and H3K9me3 also have a positive correlation with the expression of Uhrf1. However, there are a few issues. First of all, the expression level of Uhrf1 should be examined by Western. Secondly, how many times was the experiment performed, and quantification of the Western bands should be provided for statistic analysis.

We appreciate the comments. As for the original Figure S3A, we performed the Western blot for Uhrf1 and added it in new Figure S4A. Additionally, we added new data with mESCs

(Figure S4A). We performed the analysis five times for shRNA in MEFs, four times in mESCs, and quantified for the histone modification. Quantification data was presented in Figure S3A.

8. Figure 4. For Fig 4C. Please provide flow data for surface markers such as SSEA-1. Immunofluorescence images should be provided for pluripotency transcription factors (Oct4, Nanog, Sox2). For people to understand Fig 4D, it is necessary to explain the coat color and the background of the MEFs used for making the iPS cells.

We appreciate the comments. For the Figure 4C, we performed and added FACS flow data for surface marker SSEA-1. Additionally, we added immunostaining data for pluripotency markers (Oct4 and Nanog). For Figure 4D, the B6/DBA2 MEFs were used for reprogramming and coat color is agouti in white color-background of CD1 recipient. We added information of chimera formation experiment in methods section.

9. Fig S4C is not convincing. The Uhrf1 protein expression is barely detectable. What are the exact controls used in Fig 4E and Fig S4E?

We appreciate the comments. We repeated the Western blot analysis by adding more cell lysate, and present data with more convincing expression of Uhrf1 as Figure S5C.

The control for Figure 4E is the expression of Oct4, Sox2, Klf4, and Myc in wild type MEFs isolated from Uhrf1 fl/fl mouse (B6Dnk;B6N Uhrf1^{tm1a(EUCOMM)Wtsi}/leg), and the control for S4E (now, S5E) is MEFs from CD1 infected with lentivirus expressing scrambled shRNA. We changed the legend as Uhrf1 fl/fl in place of control in Figure 4E, and shScramble for Figure S5E.

10. Fig 5. The reduction in Oct4 binding could be due to a reduction in Oct4 protein expression. Oct4 protein expression in WT vs KO should be evaluated by Western. Fig 5D, it's understandable that regions that lost Oct4 binding also showed reduced chromatin accessibility, but why would regions that had replacement of Oct4 binding with Sox2 and/or Nanog binding show increased chromatin accessibility?

We appreciate the comment. We analyzed the protein expression level of OCT4 with western blotting (Figure 5A), and found that there is no difference of Oct4 level between E14 (WT) and Uhrf1 KO.

ATAC-seq reads include nucleosome-bound DNA fragment as well as transcription factor (TF)-bound DNA (Tsompana et al. Epigenetics & Chromatin, 2014). Here, we evaluated the chromatin accessibility by DNA bound by TFs (<150bp of ATAC-seq reads), including Nanog, Sox2 and Oct4. Therefore, we observed a decrease of the accessibility in loss of Oct4 binding, but not in the replacement of Oct4 binding sites, because total TF binding did not change in the replacement with Nanog or Sox2. The increase of the

accessibility was found in the replacement of Oct4 binding sites with both Nanog and Sox2 (See below). We added this figure as Figure S6B.

Reviewers' comments:

Reviewer #1 (Remarks to the Author):

The revised manuscript is improved. There are still issues that I think need to be addressed. My major concern is that the authors seem to suggest that the differentiation defects of Uhrf1 KO ES cells are mostly attributable to the Setd1a/COMPASS complex. In fact, some of the observations (e.g. cell death upon differentiation and teratoma data in Fig. 2H) can be explained by global DNA hypomethylation, because Dnmt KO ES cells show similar phenotype (see Lei et al. Development 1996; Tucker et al. PNAS 1996; Chen et al. MCB 2003). I think it's necessary to indicate the possible involvement of other Uhrf1 functions in the abstract and discussion. Another suggestion is that the authors ought to put some observations into proper context in the discussion. For example, Zscan4 and other 2C genes are only expressed in a small fraction of ES cells known as 2C-like ESCs. The downregulation of these genes could be due to reduced numbers of 2C-like ESCs, which should be indicated as a possibility in discussing the results. Also, recent work demonstrated that Zscan4 facilitates telomere elongation by downregulating Uhrf1 (Dan et al. Cell Rep 2017). Is it possible that some sort of feedback/feedforward regulation exists between Zscan4 and Uhrf1 in 2C-like ESCs? The authors indicated that Uhrf1 binding to H3R2 could enhance H3K4 methylation. Indeed, previous work has shown that H3R2 methylation (which inhibits Uhrf1 binding) and H3K4 methylation antagonize each other (Guccione et al. 2007; Kimizis et al. 2007; Hyllus et al. 2007). These papers need to be cited. There are still some typo and grammar errors (e.g. line 195, in order [other]) to; line 409, binding [biding]) that need to be corrected.

Reviewer #2 (Remarks to the Author):

While I appreciate the amount of work put into the initial submission and preparation of the revision, the authors still do not sufficiently address my major concerns related to the ChIP-seq and proteomics experiments. Rigor in these experiments is critical to support the authors' main conclusions that:

- 1) UHRF1 regulates neuroectodermal lineage (despite a title change and limited text change)
- 2) UHRF1 regulates H3K4me3 at bivalent promoters
- 3) UHRF1 interacts with the Setd1a/COMPASS complex.

The authors choose to focus on H3K4me3 as the histone mark primarily regulated by Uhrf1 at bivalent promoters, while the data from ChIP-seq experiments suggests other histone marks are equally as dynamic following Uhrf1 KO. For example, in Figure 1D, bivalent promoters in WT ESCs primarily switch to active promoters in Uhrf1 KO ESCs, i.e. H3K27me3 is lost in the absence of UHRF1 while H3K4me3 is mostly retained. This is also true in Figure S1B, where the greatest loss of histone modification peaks between WT ESCs and Uhrf1 KO ESCs is to H3K27me3 (7,486 vs 539, a loss of 93% of those peaks in KO ESCs). Furthermore, Figure 1E is important to initial observations about the neuroectodermal lineage (especially as all of Figure 2 still focuses on the neuroectodermal lineage despite a title change to the manuscript). I stand by my previous concern that without sufficient read depth, quantitative calls (changes in peak amplitude) cannot be made with this ChIP-seq data, especially as one could argue that the most striking loss of peaks is to H3K27me3 in the genes involved in muscle development.

Therefore, to support the strong statements the authors have made about H3K4me3 levels, the ChIP-seq data must be reliable and robust. According to standards listed by ENCODE (<https://www.encodeproject.org/chip-seq/histone/>), this means that for narrow-peak histone marks, which includes H3K4me3, at least 20 million reads are needed per biological replicate (two replicates required); In this dataset, only 11 million reads are made in the Uhrf1 KO ESCs for which only one replicate is presented. The same is true for H3K27me3 (8 million reads listed) and H3K9me3 (18 million reads listed), which each require a minimum of 45 million reads per

replicate. From my assessment, the number of reads and replicates given in these experiments do not reach current expected standards. Furthermore, the information cited comparing the authors' ChIP-seq data to that from ENCODE for H3K4me3 indicates comparable results for peak distribution, but is not sufficient for comparing peak amplitude or PTM abundance. Without the proper controls or normalization and proper read depth of the authors' own samples, the quantitative calls concerning loss of histone modification peaks made in this manuscript (see Figure 1E) are not rigorously supported.

At the very least, validation of ChIP results (i.e. a repeat ChIP experiment and q-PCR of a panel of loci such as in Figure 1E) and/or an orthogonal assay to confirm the conclusions is needed.

To make the conclusion that these changes to H3K4me3 are due to an interaction between Uhrf1 and Setd1a, more convincing MS data is still required. I appreciate the authors' rigor in repeating this experiment 5 times; however, I only see the results of one experiment which still suggests insignificant enrichment of components of the Setd1a/COMPASS complex. Seeing the results of all repetitions would help strengthen claims made from this data.

The authors also did not respond to my comment on Figures 4 and 5 regarding providing a stronger experimental association between Uhrf1, H3K4me3, and Oct4 binding. How do the authors speculate Uhrf1 mediates active open chromatin conformation in this situation?

I do appreciate the manner in which the authors have addressed the remainder of my concerns. The improved western blots, co-IPs, and HMT assays are more convincing and better controlled.

Finally, please note the manuscript is still difficult to read due to grammatical errors and syntax. Careful copy-editing for English language is still necessary.

Reviewer #3 (Remarks to the Author):

The authors did a great job improving the manuscript. There are just a few minor issues that need to be addressed:

Page 9 line 289-294. please re-write the confusing sentences "... However, deletion of the PHD or the TTD domains did not the HMT activity toward H3 substrate ..., while HMT activities were reduced when chromatin was used as substrate. Because chromatin substrates include DNA and core histones in addition to H3, perhaps PHD and TTD domains are essential for Uhrf1 function on chromatin, but SRA domain is essential for mediating the H3K4me3."

Neri et al. Cell 2013 and a recent paper Verma et al. Nature Genetics show that Dnmt3L and TET antagonizes DNA methylation at bivalent promoters. Can the authors speculate the connections between these studies and the roles of Uhrf1 in the current study with regard to the regulation of bivalent promoters? Although the current study doesn't focus on the regulation of DNA methylation, why are bivalent promoter regions particularly susceptible in these studies?

Overall the authors still need to improve the clarity in writing.

Response to Reviewers' comments:

We appreciate the reviewers' valuable comments and we have revised the manuscript in an effort to address comments. We think that has made the manuscript stronger.

Reviewer #1 (Remarks to the Author):

The revised manuscript is improved. There are still issues that I think need to be addressed. My major concern is that the authors seem to suggest that the differentiation defects of Uhrf1 KO ES cells are mostly attributable to the Setd1a/COMPASS complex. In fact, some of the observations (e.g. cell death upon differentiation and teratoma data in Fig. 2H) can be explained by global DNA hypomethylation, because Dnmt KO ES cells show similar phenotype (see Lei et al. Development 1996; Tucker et al. PNAS 1996; Chen et al. MCB 2003). I think it's necessary to indicate the possible involvement of other Uhrf1 functions in the abstract and discussion.

We appreciate the reviewer's insightful comment. In the revised manuscript, we have included a description of the already known mechanism of Uhrf1 and the phenotypes that are associated with DNMT1 KO mESCs and then we have broadened this to pinpoint unique differences attributed to Uhrf1 KO cells. Interestingly, in addition to cell death and teratoma size, there is overlap with respect to mesodermal differentiation in Dnmt1 KO. Similar with Uhrf1 KO EB, Dnmt1 KO EB activates mesodermal markers earlier than WT EB (Jackson et al., Mol Cell Biol, 2004). The induction of mesodermal genes was also observed by KO of PRC2 core units (EZH2 and EED) (Shen et al., Cell, 2009). Additionally, Dnmt1 plays an important role in the interplay between DNA methylation and H3K27me3 (Cooper et al., Cell Rep., 2014). These have been described in Results and Discussion sections (Line 262 – 269 in Results; Line 384 – 389 in Discussion).

However, many phenotypes are unique to loss of function of Uhrf1. Here, we demonstrated the failure of neuroectodermal specification in Uhrf1 KO cells, but has been shown that Dnmt1 KO cells can successfully differentiate into neural precursor cells (NPC) and can be induced more efficiently into astrocyte differentiation (Fan et al, Development, 2005 and Schmidt et al., PLoS One, 2012). In addition, we have demonstrated that Uhrf1 promotes iPSC reprogramming, but Dnmt1 plays a negative role in the gain of pluripotency (Mikkelsen et al. Nature 2008). Thus, we concluded that functions of Uhrf1 in neuroectodermal differentiation and iPSC reprogramming are independent of Dnmt1-related DNA hypomethylation.

Another suggestion is that the authors ought to put some observations into proper context in the discussion. For example, Zscan4 and other 2C genes are only expressed in a small fraction of ES cells known as 2C-like ESCs. The downregulation of these genes could be

due to reduced numbers of 2C-like ESCs, which should be indicated as a possibility in discussing the results. Also, recent work demonstrated that Zscan4 facilitates telomere elongation by downregulating Uhrf1 (Dan et al. Cell Rep 2017). Is it possible that some sort of feedback/feedforward regulation exists between Zscan4 and Uhrf1 in 2C-like ESCs?

We appreciate this comment. To that end we have added the description with regards to the reduction of 2C-like cells in Uhrf1 KO ESCs (Line 347 – 357 in Results, Line 424 – 432 in Discussion). In addition, we agree with reviewer’s comment about feedback regulation between Zscan4 and Uhrf1. Macfarlan et al., Nature 2012 demonstrated that 2C-like state is not stable and inter-converted with non-2C state. Therefore, the model of Dan et al. Cell Rep 2017 supports how 2C-like cells are converted back to a non-2C state, whereas our model supports how non-2C state cells are converted into 2C-like state (See below). We added the possibility of feedback regulation of Uhrf1 and Zscan4 in the Discussion section (Line 429 – 432).

The authors indicated that Uhrf1 binding to H3R2 could enhance H3K4 methylation. Indeed, previous work has shown that H3R2 methylation (which inhibits Uhrf1 binding) and H3K4 methylation antagonize each other (Guccione et al. 2007; Kimizis et al. 2007; Hyllus et al. 2007). These papers need to be cited.

We appreciate this comment and have added these citations and expanded the discussion in the manuscript to reflex the reviewers suggestion (Line 401 – 410 in Discussion).

There are still some typo and grammar errors (e.g. line 195, in order [other]) to; line 409, binding [biding]) that need to be corrected.

We have corrected the typos and grammatical errors throughout the manuscript.

Reviewer #2 (Remarks to the Author):

While I appreciate the amount of work put into the initial submission and preparation of the revision, the authors still do not sufficiently address my major concerns related to the ChIP-seq and proteomics experiments. Rigor in these experiments is critical to support the authors' main conclusions that:

- 1) UHRF1 regulates neuroectodermal lineage (despite a title change and limited text change)
- 2) UHRF1 regulates H3K4me3 at bivalent promoters
- 3) UHRF1 interacts with the Setd1a/COMPASS complex.

The authors choose to focus on H3K4me3 as the histone mark primarily regulated by Uhrf1 at bivalent promoters, while the data from ChIP-seq experiments suggests other histone marks are equally as dynamic following Uhrf1 KO. For example, in Figure 1D, bivalent promoters in WT ESCs primarily switch to active promoters in Uhrf1 KO ESCs, i.e. H3K27me3 is lost in the absence of UHRF1 while H3K4me3 is mostly retained. This is also true in Figure S1B, where the greatest loss of histone modification peaks between WT ESCs and Uhrf1 KO ESCs is to H3K27me3 (7,486 vs 539, a loss of 93% of those peaks in KO ESCs). Furthermore, Figure 1E is important to initial observations about the neuroectodermal lineage (especially as all of Figure 2 still focuses on the neuroectodermal lineage despite a title change to the manuscript).

We appreciate this constructive critique. To compare Uhrf1-mediated histone modification dynamics, we recounted ChIP-seq reads on H3K4me3 and H3K27me3 peaks (Figure S1B). The reduced level of H3K4me3 observed in Uhrf1 KO was comparable with that of H3K27me3 ($p=0.5883$ by T test, See below). Since H3K4me3 peaks are higher and sharper than H3K27me3, peak calling software still can identify peaks. Nevertheless, their heights were significantly decreased. Taken together, the disruptions of H3K4me3 and H3K27me3 occurred equally in Uhrf1 KO ESCs.

As the reviewer mentioned, loss of H3K27me3 modification on promoter regions in Uhrf1 KO is also an important finding. Therefore, in revising this manuscript, we have described the mesodermal lineage phenotype in addition to the neuroectoderm phenotype in Figure 1 and 2.

Later in the manuscript (from Figure 3), we have focused on Uhrf1-mediated H3K4me3 modification. So far, molecular functions of Uhrf1 have been extensively investigated with respect to H3K9me3 or Dnmt1 dependent mechanisms and proposed as a transcriptional suppressor or involved in the maintenance of heterochromatin. Since H3K27me3 cooperates with H3K9me3 for heterochromatin formation (Boros et al., Mol Cell Biol, 2014) and its localization to pericentric heterochromatin (PCH) is controlled by Uhrf1-mediated DNA methylation (Cooper et al., Cell Rep. 2014), it can be speculated that H3K27me3-mediated mesodermal differentiation is highly dependent on Dnmt1. Indeed, the induction of mesodermal genes after EB differentiation was also shown in Dnmt1 and PRC2 KO cells (Jackson et al., Mol Cell Biol, 2004 and Shen et al., Cell, 2009). Here we described the potential mechanism for mesoderm regulation in the discussion of our manuscript (Line 390 - 398).

In contrast to H3K27me3, the phenotypes related to H3K4me3 cannot be explained in a Dnmt1 or H3K9me3-dependent manner. As discussed in reviewer1's comment, Uhrf1 KO displayed opposing phenotypes to those observed in Dnmt1 KO with respect to neuroectodermal differentiation and pluripotency (Fan et al, Development, 2005, Schmidt et al., PLoS One, 2012 and Mikkelsen et al. Nature 2008). These results are highly indicative that Uhrf1 has alternative functions that are independent of those of Dnmt1. Furthermore, to our knowledge, the functional role of Uhrf1 in active epigenetic state has not been fully described. Thus, the investigation of Uhrf1 on active histone modification provides novel aspect of epigenetic function of Uhrf1 to the field. We have expanded the result and discussion to reflect the importance of Uhrf1-mediated H3K4me3 modifications.

I stand by my previous concern that without sufficient read depth, quantitative calls (changes in peak amplitude) cannot be made with this ChIP-seq data, especially as one

could argue that the most striking loss of peaks is to H3K27me3 in the genes involved in muscle development. Therefore, to support the strong statements the authors have made about H3K4me3 levels, the ChIP-seq data must be reliable and robust. According to standards listed by ENCODE (<https://www.encodeproject.org/chip-seq/histone/>), this means that for narrow-peak histone marks, which includes H3K4me3, at least 20 million reads are

needed per biological replicate (two replicates required); In this dataset, only 11 million reads are made in the Uhrf1 KO ESCs for which only one replicate is presented. The same is true for H3K27me3 (8 million reads listed) and H3K9me3 (18 million reads listed), which each require a minimum of 45 million reads per replicate. From my assessment, the number of reads and replicates given in these experiments do not reach current expected standards. Furthermore, the information cited comparing the authors' ChIP-seq data to that from ENCODE for H3K4me3 indicates comparable results for peak distribution, but is not sufficient for comparing peak amplitude or PTM

abundance. Without the proper controls or normalization and proper read depth of the authors' own samples, the quantitative calls concerning loss of histone modification peaks made in this manuscript (see Figure 1E) are not rigorously supported. At the very least, validation of ChIP results (i.e. a repeat ChIP experiment and q-PCR of a panel of loci such as in Figure 1E) and/or an orthogonal assay to confirm the conclusions is needed.

We appreciate the reviewer's constructive comments. We have compared the amplitude of H3K4me3 peaks ($\log_2(\text{ChIP}/\text{input})$ in H3K4me3 peaks identified by rseg software) between ours and the ENCODE ChIP-seq datasets. Similar with our previous result, peak amplitude of our ChIP-seq is strongly correlated with ENCODE, supporting the reliability of our quantification.

In addition, we have validated the histone modification changes with ChIP-qPCR in six gene loci shown in Figure 1E (new data added as Figure S1H). We found that the mesodermal markers (Hand1 and Msx2) displayed significant reductions in both H3K4me3 and H3K27me3, whereas only H3K4me3 was significantly decreased in pluripotent genes (Klf2 and Sall1) and neuroectodermal genes (Nes and Tubb3). This is consistent with our global analysis of ChIP-seq (Figure S1G). Overall, these results further support our ChIP-seq results. We hope that this appeases the reviewers concerns.

To make the conclusion that these changes to H3K4me3 are due to an interaction between Uhrf1 and Setd1a, more convincing MS data is still required. I appreciate the authors' rigor in repeating this experiment 5 times; however, I only see the results of one experiment which still suggests insignificant enrichment of components of the Setd1a/COMPASS complex. Seeing the results of all repetitions would help strengthen claims made from this data.

We are sorry about the confusion caused. In the previous Table S1, we presented total

amount of peptides obtained from five MS experiments. In new Table S1, we added the number of peptides in each MS experiment. We note that Uhrf1 was identified in all five replicates. However, because the peptide count in each replicate is very limited, the enrichment of Setd1a/COMPASS and the known Uhrf1-interacting proteins was not always identified in replicates. This may be due to the nature of the protein that the interaction with the associated proteins may be weak or unstable. Therefore, to provide confidence in the proposed interaction, we performed Co-IP experiments and *in vitro* binding assay with purified recombinant protein from *E.coli* (Uhrf1) and insect cells (SETD1A). Importantly, we clearly demonstrated their direct interaction (Figure S4G). Furthermore, using the deletion mutants we mapped the interacting domains of Uhrf1 and Setd1a (Figure 3D, 3E, and S4C).

The authors also did not respond to my comment on Figures 4 and 5 regarding providing a stronger experimental association between Uhrf1, H3K4me3, and Oct4 binding. How do the authors speculate Uhrf1 mediates active open chromatin conformation in this situation?

We now show that Uhrf1 mediates H3K4me3 modification with Set1a/COMPASS (Figure 3). In addition, we also validated that Uhrf1 interacts with Oct4 (Figure 5A). Our ATAC-seq and ChIP-seq analyses revealed that Uhrf1 is enriched on Oct4 binding sites and that Uhrf1 KO significantly reduced H3K4me3 on Oct4 binding sites (Figure 5D-F). We believed that these results sufficiently support our hypothesis that Uhrf1 mediates active open chromatin conformation on Oct4 binding sites.

Fang et al., Stem Cell, 2016, demonstrated that Set1a promotes iPSC reprogramming of MEFs, which is the same phenotype observed with Uhrf1 (Figure 4). Set1a is co-localized with Oct4 and co-activates downstream genes by mediating H3K4me3. In addition, other works (Macfarlan et al., Nature 2012 and Genes Dev 2011) showed that active histone marks including H3K4me3 are required for activation of 2C-specific genes and MERVL. These previous findings further support our conclusion, and we added these to discussion in the manuscript.

As core factors of Uhrf1-mediated iPSC reprogramming, we focused on 2C-related genes (Figure 5G-H and S6G-H). To further validate active chromatin conformation in these loci, we performed ChIP-qPCR for H3K4me3 and H3K27me3. Despite no significant difference in H3K27me3, H3K4me3 was dramatically decreased in the Uhrf1 KO. Overall, Uhrf1-mediated active open chromatin conformation was strongly supported in the manuscript. We added ChIP-qPCR results as Figure S6E-F. We hope this addresses the reviewers concerns.

I do appreciate the manner in which the authors have addressed the remainder of my concerns. The improved western blots, co-IPs, and HMT assays are more convincing and better controlled.

Finally, please note the manuscript is still difficult to read due to grammatical errors and

syntax. Careful copy-editing for English language is still necessary.

We corrected and addressed the grammatical issues.

Reviewer #3 (Remarks to the Author):

The authors did a great job improving the manuscript. There are just a few minor issues that need to be addressed:

Page 9 line 289-294. please re-write the confusing sentences "... However, deletion of the PHD or the TTD domains did not the HMT activity toward H3 substrate ..., while HMT activities were reduced when chromatin was used as substrate. Because chromatin substrates include DNA and core histones in addition to H3, perhaps PHD and TTD domains are essential for Uhrf1 function on chromatin, but SRA domain is essential for mediating the H3K4me3."

We really appreciate reviewer's comment. We have rewritten this sentence to add clarity. Please see the Line 293 – 301).

Neri et al. Cell 2013 and a recent paper Verma et al. Nature Genetics show that Dnmt3L and TET antagonizes DNA methylation at bivalent promoters. Can the authors speculate the connections between these studies and the roles of Uhrf1 in the current study with regard to the regulation of bivalent promoters? Although the current study doesn't focus on the regulation of DNA methylation, why are bivalent promoter regions particularly susceptible in these studies?

We appreciate this important suggestion. We have carefully considered the effect of TET and Dnmt3L proteins on Uhrf1 binding on bivalent promoter. Using public ChIP-seq for TET1, TET2 and DNMT3L, the co-localization of these proteins in comparison with Uhrf1 were analyzed. Although 500 - 2,000 peaks showed the co-localization with Uhrf1, most of peaks did not show significant enrichment for Uhrf1 (Figure below). TET and DNMT3L may also bound non-bivalent regions where Uhrf1 was not enriched. In addition, our MS data did not indicate any significant interaction of Uhrf1 with either TET or DNMT3L. Thus, it is difficult to connect with our model the antagonizing effect of TET and DNMT3L on Uhrf1. However, it is important to discuss how bivalent promoters are susceptible by functional loss of Uhrf1. Rajakumara et al. previously demonstrated Uhrf1 preferentially binds with unmodified H3R2, which antagonizes H3K4 methylation. Therefore, some of histone modifications specific to bivalent regions or their writer, reader and eraser proteins may be

candidates to control Uhrf1 localization. Therefore we have expanded our discussion on this (Line 401- 410) in the revised manuscript.

Overall the authors still need to improve the clarity in writing.

We have addressed this and improved the writing.

REVIEWERS' COMMENTS:

Reviewer #1 (Remarks to the Author):

The issues and concerns raised by me have been satisfactorily addressed in the revised manuscript.

Reviewer #2 (Remarks to the Author):

I appreciate the authors' response to my requests. New data is helpful in supporting the conclusions made in the manuscript.

The authors note that upon "recounting ChIP-seq reads" (Fig S1B), H3K4me3 and H3K27me3 reductions occur similarly (as one would predict from their other data and figures), and the authors added additional language exploring the loss of not just H3K4me3 at bivalent promoters but also H3K27me3 and its effect on the mesodermal lineage. While it would be nice to have experimental support for the speculated suggestion that the loss of H3K27me3 and induction of mesodermal genes is due to failed Dnmt1-mediated recruitment of PRC2 in the absence of Uhrf1 (lines 261-268 and 390-398), I appreciate the added text giving at least some attention to this potential mechanism.

It still appears the ChIP-seq data does not fully meet ENCODE guidelines concerning biological duplicates, read-depth, and sufficient input, while the figure in the rebuttal comparing peaks of the authors' H3K4me3 data set to ENCODE does not seem to have nearly as strong of a correlation as the two Encode sets to each other. That being said, I do appreciate the candidate-level validation shown in Fig S1H of the ChIP data in Figure 1E. While the validation of Msx2 does not appear to closely match the genome browser shot (which does not indicate a loss of H3K4me3 in contrast to the ChIP-qPCR of Fig S1H), the validation of the other five genes appears to be consistent with ChIP-seq. Also, is "Hand1" in S1H supposed to be "Hand2" as in Fig 1E?

Thank you for the clarification of the mass spec data and the number of replicates. As the presented numbers represent the aggregate of the five different experiments, it shows the MS data is of poor quality and should be removed from the manuscript. Uhrf1 itself should have much higher counts than it does either total or in each of the replicates (and, as Uhrf1 is the bait, this should be observed regardless of the proposed "weak or unstable" interactions). Furthermore, the components of Setd1a/COMPASS are not consistently represented across replicates. The IP experiments, however, are more convincing than before and lend credence to the proposed Uhrf1-Setd1a/COMPASS interaction.

REVIEWERS' COMMENTS:

Reviewer #1 (Remarks to the Author):

The issues and concerns raised by me have been satisfactorily addressed in the revised manuscript.

We appreciate meaningful reviewer's comment.

Reviewer #2 (Remarks to the Author):

I appreciate the authors' response to my requests. New data is helpful in supporting the conclusions made in the manuscript.

The authors note that upon "recounting ChIP-seq reads" (Fig S1B), H3K4me3 and H3K27me3 reductions occur similarly (as one would predict from their other data and figures), and the authors added additional language exploring the loss of not just H3K4me3 at bivalent promoters but also H3K27me3 and its effect on the mesodermal lineage. While it would be nice to have experimental support for the speculated suggestion that the loss of H3K27me3 and induction of mesodermal genes is due to failed Dnmt1-mediated recruitment of PRC2 in the absence of Uhrf1 (lines 261-268 and 390-398), I appreciate the added text giving at least some attention to this potential mechanism.

We appreciate for valuable comment to improve our manuscript.

It still appears the ChIP-seq data does not fully meet ENCODE guidelines concerning biological duplicates, read-depth, and sufficient input, while the figure in the rebuttal comparing peaks of the authors' H3K4me3 data set to ENCODE does not seem to have nearly as strong of a correlation as the two Encode sets to each other. That being said, I do appreciate the candidate-level validation shown in Fig S1H of the ChIP data in Figure 1E. While the validation of Msx2 does not appear to closely match the genome browser shot (which does not indicate a loss of H3K4me3 in contrast to the ChIP-qPCR of Fig S1H), the validation of the other five genes appears to be consistent with ChIP-seq. Also, is "Hand1" in S1H supposed to be "Hand2" as in Fig 1E?

Thank you for great discussion. We designed ChIP-qPCR primers for intron 1 region of *Msx2* (read arrow in below). In this region, we observed reduction of both H3K4me3 and H3K27me3 in ChIP-seq, which are consistent with our ChIP-qPCR data (Fig. S1H).

We corrected “Hand1” as “Hand2”.

Thank you for the clarification of the mass spec data and the number of replicates. As the presented numbers represent the aggregate of the five different experiments, it shows the MS data is of poor quality and should be removed from the manuscript. Uhrf1 itself should have much higher counts than it does either total or in each of the replicates (and, as Uhrf1 is the bait, this should be observed regardless of the proposed “weak or unstable” interactions). Furthermore, the components of Setd1a/COMPASS are not consistently represented across replicates. The IP experiments, however, are more convincing than before and lend credence to the proposed Uhrf1-Setd1a/COMPASS interaction.

We carefully considered whether we should include or not our proteomics data in the manuscript. For the purification of Uhrf1-interacting protein, we used the affinity capture by biotinylation-streptavidin (bio-SA) association. This is a helpful for protein pulldown independently from antibody quality (Wang et al., Nature, 2006). Bio-SA pulldown may also capture background proteins with endogenous biotinylated proteins, which are involved in mRNA and ribosome processing, but rare in transcriptional regulation (de Boer et al., PNAS, 2003). Therefore, when we select or rank Uhrf1-interacting proteins, those enriched in BirA control (potentially-biotinylated proteins) must be removed. Indeed, Mdn1 and Ncl, which showed the top highest peptide counts in BirA control, are involved in ribosomal synthesis and maturation. As we discussed previously, normalized count ($\log_2(\text{ratio})$) is more reliable measure for Uhrf1 interaction than actual peptide count, and Uhrf1 was ranked as top by this criteria (Table S1). Lower counts of Setd1a/COMPASS complex peptide may be caused by higher expression of background proteins. The reduction of background proteins in future proteomics should be improved. However, we performed Co-IP experiment as suggested by reviewer to validate proteomics-derived Uhrf1 interaction and obtained highly consistent Uhrf1-interaction by Co-IP experiment. Overall, all these results strongly support the reliability of our proteomics data, supporting the retention of our proteomics result in the manuscript. However, because the proteomics data is not of high quality, we have moved the data as supplementary data as suggested by editor.